# HiTE: a fast and accurate dynamic boundary adjustment approach for full-length transposable element detection and annotation

Kang Hu[1,2,3,10], Peng Ni [1,2,3,10], Minghua Xu[1,3], You Zou[1,3], Jianye Chang[4], Xin Gao [5,6], Yaohang Li [7], Jue Ruan [4], Bin Hu[8,9] ✉ & Jianxin Wang [1,2,3] ✉

Recent advancements in genome assembly have greatly improved the prospects for comprehensive annotation of Transposable Elements (TEs). However, existing methods for TE annotation using genome assemblies suffer from limited accuracy and robustness, requiring extensive manual editing. In addition, the currently available gold-standard TE databases are not comprehensive, even for extensively studied species, highlighting the critical need for an automated TE detection method to supplement existing repositories. In this study, we introduce HiTE, a fast and accurate dynamic boundary adjustment approach designed to detect full-length TEs. The experimental results demonstrate that HiTE outperforms RepeatModeler2, the state-of-the-art tool, across various species. Furthermore, HiTE has identified numerous novel transposons with well-defined structures containing protein-coding domains, some of which are directly inserted within crucial genes, leading to direct alterations in gene expression. A Nextflow version of HiTE is also available, with enhanced parallelism, reproducibility, and portability.

Transposable elements (TEs), which make up the majority of repetitive regions in most eukaryotic species[1–3], are known to have a significant impact on genome evolution and intraspecific genomic diversity[4,5]. TEs have been found to play a key role in human diseases and crop breeding by interrupting or regulating the key genes[6–8]. Identifying intact TEs is challenging due to various complications[9], including but not limited to: (i) the varying degradation rates of TEs, which can lead to the loss of structural signals[10]; (ii) the complex pattern of TE sequences, resulting from random deletions, insertions[11,12], and nested TE[13]; (iii) the difficulty to determine the true ends of highly fragmented TE instances[14]; (iv) the obstacle to construct full-length TE models posed by the abundance of fragmented TEs; (v) the confounding impact of regional homology between unrelated TEs on their identification and classification; and (vi) the risk of erroneously identifying

[1]School of Computer Science and Engineering, Central South University, Changsha 410083, China. [2]Xiangjiang Laboratory, Changsha 410205, China. [3]Hunan Provincial Key Lab on Bioinformatics, Central South University, Changsha 410083, China. [4]Shenzhen Branch, Guangdong Laboratory for Lingnan Modern Agriculture, Genome Analysis Laboratory of the Ministry of Agriculture and Rural Affairs, Agricultural Genomics Institute at Shenzhen, Chinese Academy of Agricultural Sciences, Shenzhen 518000, China. [5]Computer Science Program, Computer, Electrical and Mathematical Sciences and Engineering Division, King Abdullah University of Science and Technology (KAUST), Thuwal, Saudi Arabia. [6]Center of Excellence on Smart Health, King Abdullah University of Science and Technology (KAUST), Thuwal, Saudi Arabia. [7]Department of Computer Science, Old Dominion University, Norfolk, VA 23529, USA. [8]Key Laboratory of Brain Health Intelligent Evaluation and Intervention, Ministry of Education (Beijing Institute of Technology), Beijing, P. R. China. [9]School of Medical Technology, Beijing Institute of Technology, Beijing, P. R. China. [10]These authors contributed equally: Kang Hu, Peng Ni. ✉e-mail: bh@bit.edu.cn; jxwang@mail.csu.edu.cn

high copy numbers of segmental duplications or tandem repeats as putative TE instances.

There are several tools available for the automated identification and annotation of TEs, which can be broadly divided into three categories: (i) De novo methods, (ii) Signature-based methods, and (iii) TE discovery pipelines. By identifying exact or closely matching repetitions, de novo methods, which mainly include a (spaced) k-mer-based or self-comparison approach, can identify novel TE instances that do not belong to a known family of TE. Although k-mer-based approaches, such as RepeatScout[15] and P-Clouds[16], are better suited for dealing with young TEs with plenty of copies, they may produce highly fragmented sequences for older TEs with diverse or complex patterns. On the other hand, self-comparison methods, such as Grouper[17], RECON[18], and PILER[19], can identify more sophisticated TE families using intensive and sensitive alignments, but accurately clustering highly fragmented and mosaic TE sequences remains challenging. Signature-based methods, such as LTRharvest[20], LTR_retriever[21], Generic Repeat Finder[22], EAHelitron[23], HelitronScanner[24], and MITE-Hunter[25], identify TEs based on family-specific features. These methods can overcome the limitations of purely de novo methods that may miss well-characterized TEs with low copy numbers. However, signature-based methods are prone to false positives due to the weak structural characteristics of many TEs. TE discovery pipelines, like EDTA[26], RepeatModeler2[27], and EarlGrey[28], combine different TE identification tools to comprehensively identify all types of TEs within a given genome. While these pipelines can overcome the limitations of individual tools, they also introduce their inherent defects and require careful handling of redundant results. A high-quality TE library can help overcome these challenges by providing a comprehensive and structured collection of TEs. After years of manual curation, Repbase[29] and Dfam[30] are high-quality consensus libraries for a limited set of species, nevertheless, all automatically generated TE libraries still require extensive manual editing[31]. An ideal TE library should only contain full-length TE models[9]. The inclusion of fragmented sequences in the TE library would hamper the classification of the TE families, inflate the number of actual TE families in the genome, and confound genome annotation and downstream analyses[27]. However, the reality is that almost all tools will inevitably introduce some fragmented TE models. RepeatModeler2 introduces a benchmarking method to quantify full-length and fragmented TE models. Specifically, it uses *Perfect* indicators to represent full-length TE models, while *Good* and *Present* indicators represent fragmented TE models.

In this study, we introduce HiTE, a fast and accurate dynamic boundary adjustment method for detecting full-length TEs with high precision. For example, when detecting transposons in the model species rice, EDTA and RepeatModeler2 take around 10 and 26 h, respectively. In contrast, HiTE accomplishes the task in 2 h, resulting in time savings of approximately 5-fold and 13-fold, respectively. Using highly conservative structural features and multiple copies, HiTE discovers many novel TE instances and produces a high-quality, structurally intact, and classified TE library. We further develop a Nextflow[32] pipeline of HiTE, which enhances the reproducibility and portability of HiTE. Comprehensive experiments on nine species, ranging from plants to animals, show that HiTE outperforms other tools in terms of accuracy and the capability of identifying full-length TE models. The results of this study demonstrate the potential of HiTE for the creation of a high-quality TE library for a wide range of species.

## Results

### Overview of HiTE

Purely de novo methods for detecting TEs based on sequence repetition alone may miss low-copy but well-characterized instances, whereas signature-based methods are susceptible to false positives, owing to the poor structural characteristics of certain TEs[9,33]. HiTE is an automated TE annotation pipeline that combines the strengths of de novo and signature-based methods, aiming to produce a high-quality TE library. To achieve comprehensive TE annotation, HiTE develops various useful algorithms and does not require manual intervention. Considering the diverse structural characteristics and distribution of TEs in the genome, HiTE primarily consists of four modules, namely structural-based LTR (Long Terminal Repeat) searching (Fig. 1h), homology-based non-LTR searching (Fig. 1i), de novo TE searching (Fig. 1d), and signature-based TE searching (Fig. 1e). These modules can identify nearly all types of transposons, including LTR, TIR (Terminal Inverted Repeat), Helitron, LINE (Long Interspersed Nuclear Element), and SINE (Short Interspersed Nuclear Element)[34].

Firstly, to reduce computational cost during a single run, HiTE initially partitions the genome assembly into smaller chunks and applies Tandem Repeats Finder (TRF)[35] to mask tandem repeats (Fig. 1b, c). To minimize redundant computations in large genomes, we implement a "mask-identify-mask..." computation method. Before iteratively detecting TEs for each new genome chunk, we first mask the full-length copies of identified TEs in the chunk. Given the repetitive nature of TEs and their numerous copies in the genome, this method significantly reduces redundant computations. Secondly, we have devised HiTE-FMEA, a module that uses sensitive alignment algorithms to identify TE models with coarse-grained boundaries (Fig. 1d, Methods). Unlike RECON and RepeatScout, which focus on achieving precise TE boundaries, HiTE-FMEA allows alignment errors to connect and span insertions and gaps between alignment segments, aiming to preserve the integrity of TE instances as much as possible (Supplementary Note 5). The FEMA algorithm is the de novo TE searching module of HiTE, which provides the input for the subsequent signature-based TE searching module (Fig. 1e).

Then, we have developed HiTE-TIR, HiTE-Helitron, and HiTE-NonLTR modules to detect TIR, Helitron, and non-LTR transposons with fine-grained boundaries (Fig. 1e, Methods). As shown in Supplementary Fig. 2b, given the misleading nature of TE structural signals, the signature-based methods using the whole genome as input often lead to more false positives. Therefore, HiTE performs identification on all potential TE candidate fragments generated by our FMEA algorithm. For example, we search for specific structural features of TEs, such as target site duplications (TSDs), terminal inverted repeats, and hairpin loops, within the coarse-grained TE candidates identified by HiTE-FMEA. To minimize false positives, such as segmental duplications, we have also implemented multiple filtering methods, including a novel homology-based filtering method based on multiple sequence alignment (Supplementary Figs. 4 and 5). This method not only filters out many false-positive sequences but also dynamically identifies the genuine TE boundaries, thereby further refining the boundaries of TE instances (Supplementary Note 3).

Next, we use LTR_FINDER[36] and LTRharvest[20] to identify all candidate LTR-RTs (LTR retrotransposons), which are then subjected to stringent filtering using LTR_retriever[21] to identify reliable LTR-RTs (Fig. 1h, Supplementary Fig. 6). Due to the presence of deletions in LTR transposons, existing clustering methods like CD-HIT-EST struggle with redundancy removal. To solve this problem, HiTE begins with BLASTN for all-vs-all comparisons, applies our FMEA algorithm to bridge gaps, clusters sequences that bridge these gaps, conducts multiple sequence alignment using Mafft[37,38], and generates consensus sequences based on the majority rule (Methods). In addition, due to the variability and lack of discernible structural signals of non-LTR elements, we have developed a homology-based non-LTR searching module called HiTE-NonLTR-homology to achieve high-precision non-LTR annotation (Fig. 1i, Supplementary Tables 9-11). Finally, we collect all types of TEs together to generate an intact TE consensus library (Fig. 1g, Methods). HiTE offers two approaches for TE classification: RepeatClassifier and NeuralTE[39]. RepeatClassifier, implemented within RepeatModeler2, employs homology-based searches to classify TEs with sequence similarity to known transposons. On the other hand,

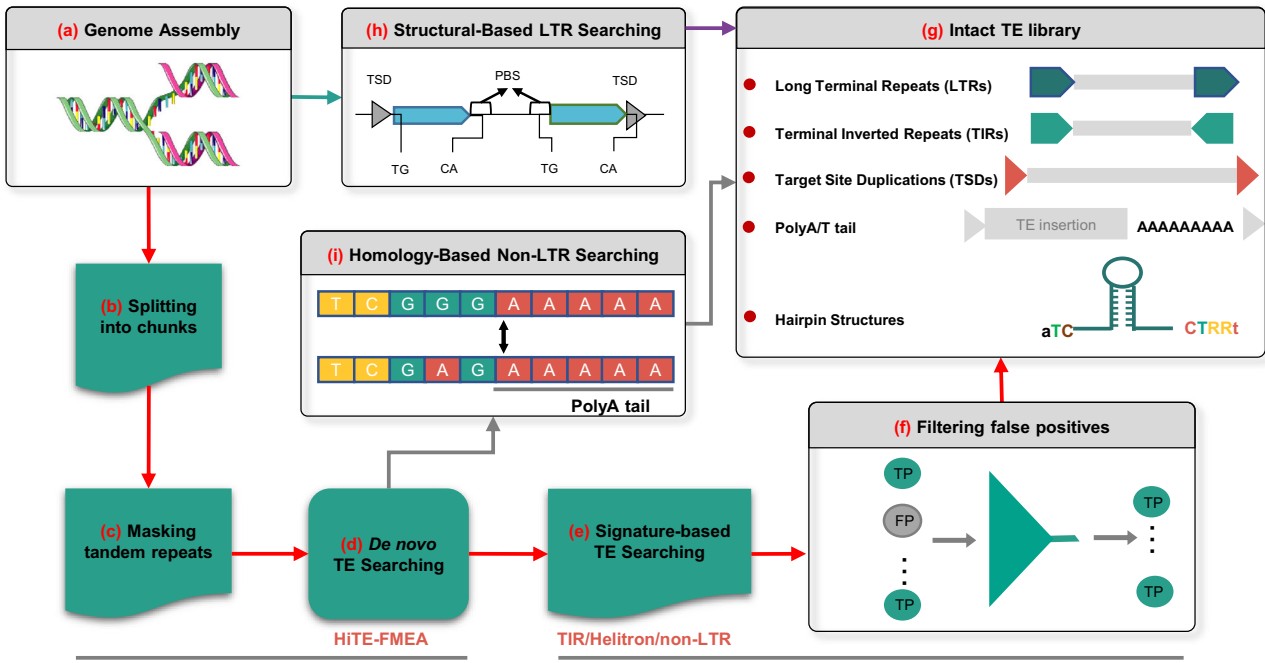

**Fig. 1 | Workflow of the HiTE pipeline for TE annotation. a** HiTE takes the genome assembly as input. **b** The genome assembly is split into chunks to reduce the amount of single-round computation. **c** Sequences containing tandem repeats are filtered out as false positives. **d**, **h**, and **i** Three modules of HiTE, De novo TE searching, Structural-based LTR searching, and Homology-based non-LTR searching, are developed to identify different types of TEs. **e** Signature-based TE searching module is used to identify TIR, Helitron, and non-LTR elements with fine-grained boundaries. **f** False positives are filtered out using reliable strategies, including a novel homology-based filtering method based on multiple sequence alignment. **g** The classified TE libraries generated by HiTE have intact TE structures.

NeuralTE uses a deep learning method to achieve superior classification for novel transposons by identifying various structural features of transposons. Users are recommended to use RepeatClassifier when dealing with highly studied genomes and to use NeuralTE in other cases. We have outlined the main differences between HiTE and existing tools in Supplementary Note 6.

## HiTE accurately detects more intact TE models

A well-established evaluation method is crucial for accurately assessing the performance of different methods in a competitive manner[9]. However, a standardized benchmarking method has not been proposed. Compared to various evaluation methods, two benchmarking methods introduced in recent studies, namely RepeatModeler2 and EDTA[26], have been shown to be more reasonable (referred to as BM_RM2 and BM_EDTA hereafter). BM_RM2 is based on RepeatMasker (4.1.1) and a custom bash script (https://github.com/jmf422/TE_annotation/blob/master/get_family_summary_paper.sh) provided by RepeatModeler2. BM_EDTA is based on the Perl script "lib-test.pl" included in EDTA. An ideal TE library should only contain full-length TE models, which can be evaluated by the *Perfect* indicator from BM_RM2. TE models are considered *Perfect* matches with >95% sequence similarity, >95% length coverage, and <5% divergence for a family consensus in the gold standard library. Given that the protein sequence required for transposition is contained within the full-length TE, the number of *Perfect* models is the most meaningful metric for evaluating TE integrity and biological significance[27,40].

BM_EDTA involves comparing the test annotation to existing curated genome annotations and calculating true positives (TP) and false negatives (FN). As shown in Supplementary Table 3, HiTE shows the highest precision compared to existing tools in the BM_EDTA evaluation. Due to the exclusion of numerous fragmented sequences lacking clear structures or multiple-copy support, HiTE exhibits relatively lower sensitivity compared to other tools using BM_EDTA.

However, the fragmented TE families can result in higher sensitivity when using BM_EDTA (Supplementary Note 2). This indicates that BM_EDTA cannot exclude the influence from fragmented sequences, potentially leading to misleadingly high sensitivity.

To better assess the quality of the test library, we have designed a new evaluation approach, referred to as BM_HiTE, as shown in Supplementary Fig. 1. The major difference between BM_HiTE and BM_EDTA lies in the criteria for calculating true positives. BM_EDTA considers any sequence that has any length of match with the standard library as a true positive. In BM_HiTE, a sequence is considered a true positive only when the length of the overlap between the test and Repbase TE sequence exceeds a threshold, such as 95% of their lengths respectively. All others are considered as false positives, including alignments with significant shifts, longer sequences containing the true TE, and fragmented TE sequences (Supplementary Fig. 1). BM_HiTE mitigates the influence of fragmented sequences on the results, making it more suitable for evaluating high-quality full-length TE libraries. BM_HiTE defaults to using a 95% length coverage threshold, drawing inspiration from BM_RM2. However, such a threshold is still considered inaccurate and misleading[31]. Therefore, we have used two distinct thresholds, namely the rough perfect definition (greater than 95% length coverage) and the precise perfect definition (greater than 99% length coverage), to assess the performance of different tools. BM_HiTE also supports user-defined length coverage thresholds. Considering the traditional 80-80-80 rule for consensus sequence generation, we have incorporated performance evaluations of various tools based on the 80% length coverage.

HiTE demonstrates superior overall performance compared to existing tools, exhibiting greater stability across all species when using various thresholds (Fig. 2, Supplementary Table 2). For example, when transitioning from the rough perfect definition to the precise perfect

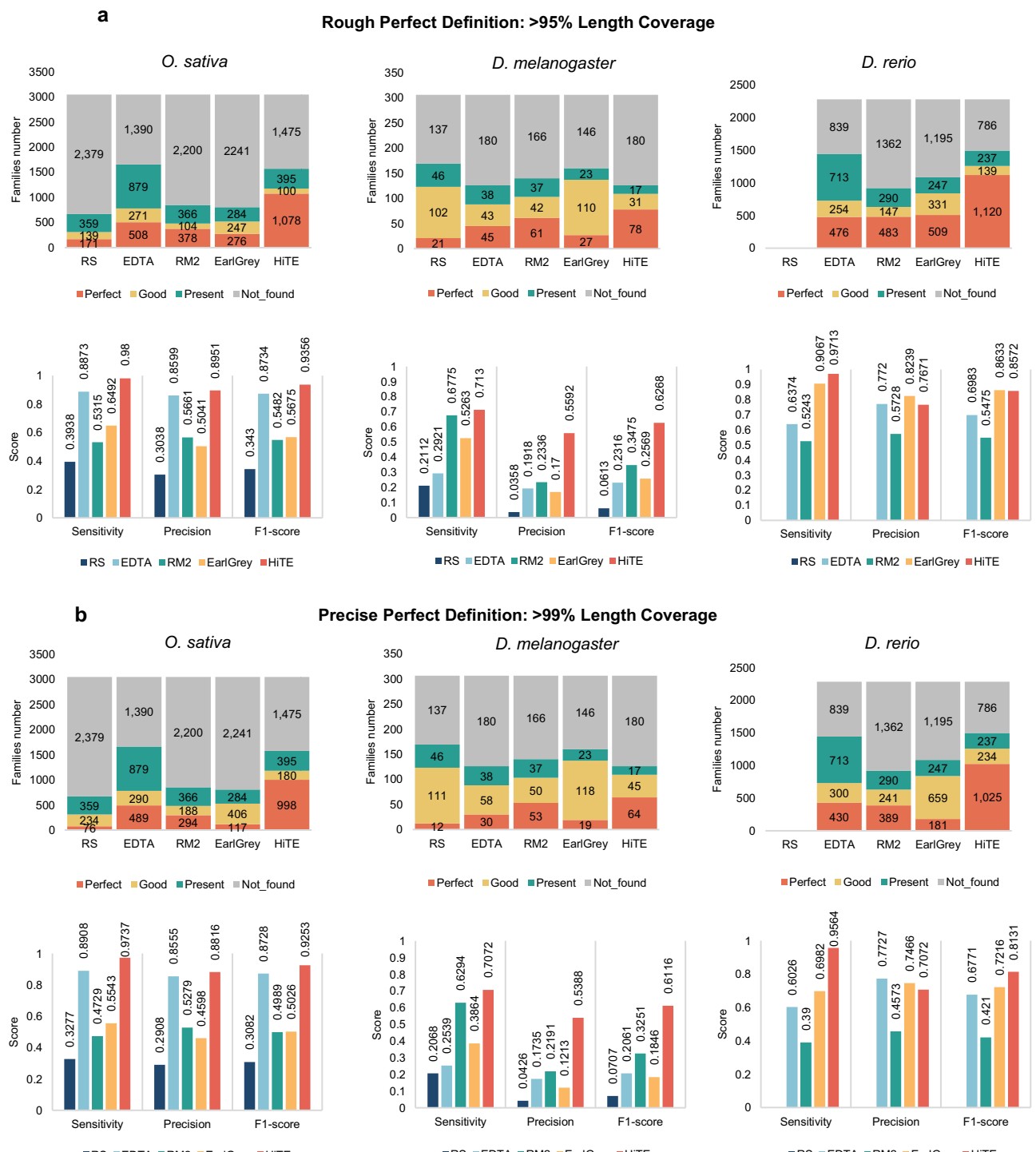

**Fig. 2 | Evaluation of HiTE for accurately identifying structurally intact TEs using BM_RM2 and BM_HiTE. a, b** Performance comparison of different tools based on rough and precise perfect definitions, respectively. Due to the limitation of RepeatScout in handling genomes larger than 1 GB, its results are unavailable for *D. rerio*. RS: RepeatScout; RM2: RepeatModeler2.

definition, the number of perfect TE families identified by HiTE only slightly decreases from 1078 to 998 in *O. sativa*, and from 1120 to 1025 in *D. rerio*. This suggests that HiTE identifies TE families with accurate boundaries, which can reduce the requirement for extensive manual curation when building a curated library. HiTE also outperforms the other four tools across most species, exhibiting higher precision and sensitivity at nearly all thresholds (Fig. 2, Supplementary Table 4). The exception is seen in *M. musculus*, where we find that EarlGrey shows higher performance than HiTE using BM_HiTE. Although HiTE achieves

the highest performance at the 80% coverage threshold on *M. musculus*, EarlGrey obtains higher performance at 95% and 99% thresholds (Supplementary Table 4). However, due to the unstable performance of EarlGrey across different species and its longer runtime (Supplementary Table 13), it cannot meet the increasing demand for TE identification and annotation in non-model species.

To gain a deeper understanding of the distinctions between the TE library produced by HiTE and those from other tools, Supplementary Note 1 offers a comprehensive analysis, providing detailed insights

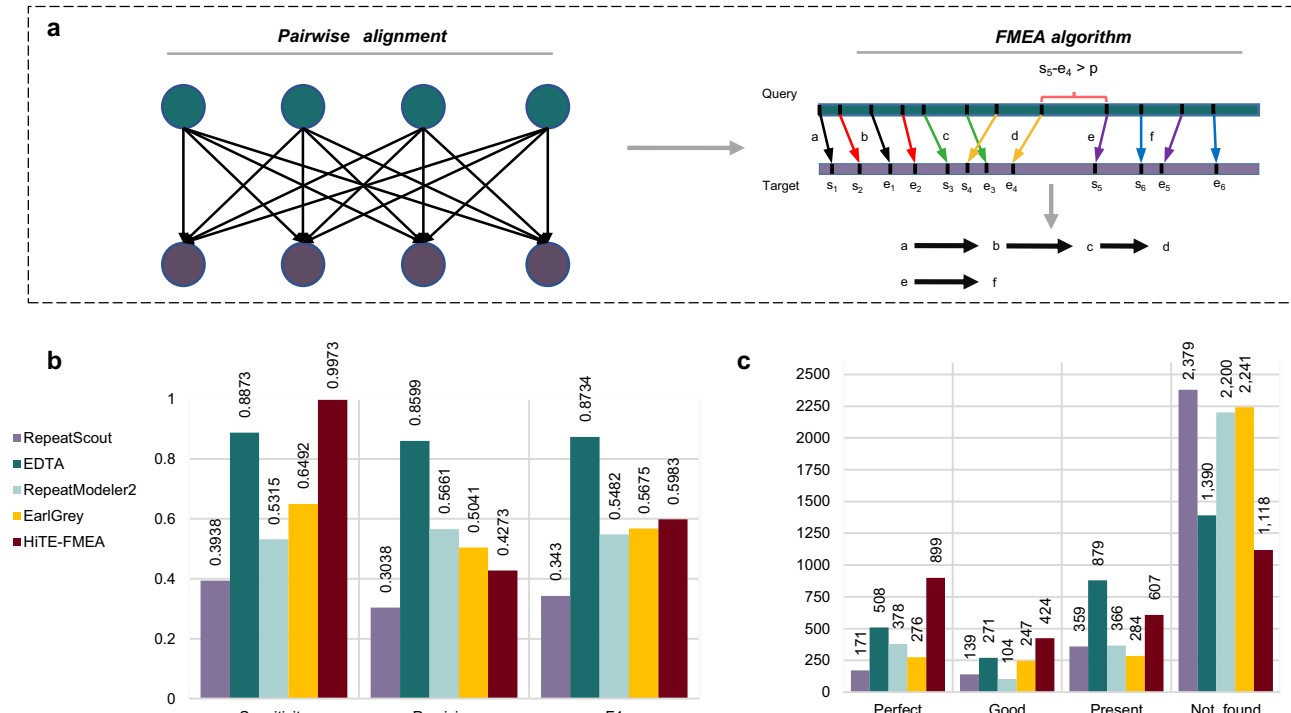

**Fig. 3 | Evaluation of HiTE-FMEA for TE annotation based on *O. sativa*.**
**a** Illustration of pairwise alignments of genome assembly contigs and the fault-tolerant mapping expansion algorithm, FMEA. Two TE instances included in a contig sequence, a-d and e-f, are identified by FMEA. **b, c** Evaluation of HiTE-FMEA against RepeatScout, EDTA, RepeatModeler2, and EarlGrey using BM_HiTE and BM_RM2 under the rough perfect definition, respectively.

and evaluations in comparison to widely used TE libraries. In summary, HiTE shows higher performance compared to the other four tools across nearly all datasets (Supplementary Tables 2 and 4).

## HiTE-FMEA detects TE models with coarse-grained boundaries

De novo methods based on identifying exact or closely matching repetitions can be used to discover novel instances that do not belong to the known TE families. Existing methods (such as RECON) cluster fragmented sequences into piles based on alignment overlap relationships and identify TE families using strategies like single linkage clustering. Due to the differentiation caused by mutations (snp, indel, etc.), many fragmented alignments are generated, making it challenging for existing clustering methods to correctly identify the true boundaries of TEs. To address this issue, we have developed a fault-tolerant mapping expansion algorithm (FMEA). Instead of directly clustering fragmented sequences, FMEA attempts to bridge and traverse some insertions and gaps between alignment segments, thus preserving the integrity of the TE structures (Fig. 3a, Methods).

As shown in Fig. 3c, compared to RepeatScout, EDTA, Repeat-Modeler2, and EarlGrey, HiTE-FMEA can retain more full-length TE families (*Perfect* category). Nevertheless, HiTE-FMEA can only identify TE models with coarse-grained boundaries, leading to potential fragments and ambiguous ends of candidates. As shown in Fig. 3b, when using BM_HiTE, HiTE-FMEA achieves the highest sensitivity but the lowest precision, indicating the presence of false-positive fragments. A similar observation is noted when using BM_EDTA (Supplementary Table 3). Therefore, we need to further develop signature-based and filtering methods to ensure the reliability of the identification results.

Upon manual inspection, the sequences identified by HiTE-FMEA have been found to include not only transposons but also non-TEs, such as segmental duplications and multi-copy genes. While HiTE-FMEA may not be able to discover the exact true ends of TE families, it identifies the most *Perfect* TE models when compared to the other four tools (Fig. 3c). Specifically, HiTE-FMEA has identified 899 *Perfect* TE

families, which is higher (by 426%, 75%, 138%, and 226%) than the corresponding families identified by RepeatScout, EDTA, RepeatModeler2, and EarlGrey, respectively. However, HiTE-FMEA has also identified 424 *Good* and 607 *Present* TE families, suggesting the existence of fragmented TE sequences that necessitate further processing.

## HiTE detects TE models with fine-grained boundaries

HiTE-FMEA shows limitations in accurately identifying TEs. At the same time, identifying transposable elements, such as Terminal inverted repeat (TIR)[41], Helitron[42] and non-LTR elements, is challenging due to their weak structural signals. To address this issue, we design signature-based identification methods for TIR, Helitron, and non-LTR elements, called HiTE-TIR, HiTE-Helitron, and HiTE-NonLTR (Fig. 4a, Methods), which can produce high-precision and structurally intact TE models. In contrast to existing methods that use the whole genome as input, the signature-based approach of HiTE performs identification on the potential TE candidate fragments generated by HiTE-FMEA module. As shown in Supplementary Fig. 2b, given the misleading nature of TE structural signals, the signature-based methods using the whole genome as input often lead to more false positives.

As shown in Supplementary Table 5, under the same conditions without using filtering algorithms, the signature-based method of HiTE (HiTE-NoFiltering) identifies more full-length TE families compared to the signature-based method of EDTA (EDTA-NoFiltering). For instance, HiTE-NoFiltering identifies 613 perfect TIR families and 104 Helitron families in *O. sativa*, whereas EDTA-NoFiltering only identifies 208 and 3, respectively. When using BM_EDTA, HiTE-NoFiltering shows comparable or higher performance compared to EDTA-NoFiltering (Supplementary Table 6).

To reduce false positives and generate the cleanest possible library, we have developed a novel homology-based filtering method based on multiple sequence alignment (Fig. 4a, Methods). The key advantage of the filtering method lies in its ability to not only filter out false positives but also dynamically identify TE boundaries, allowing us

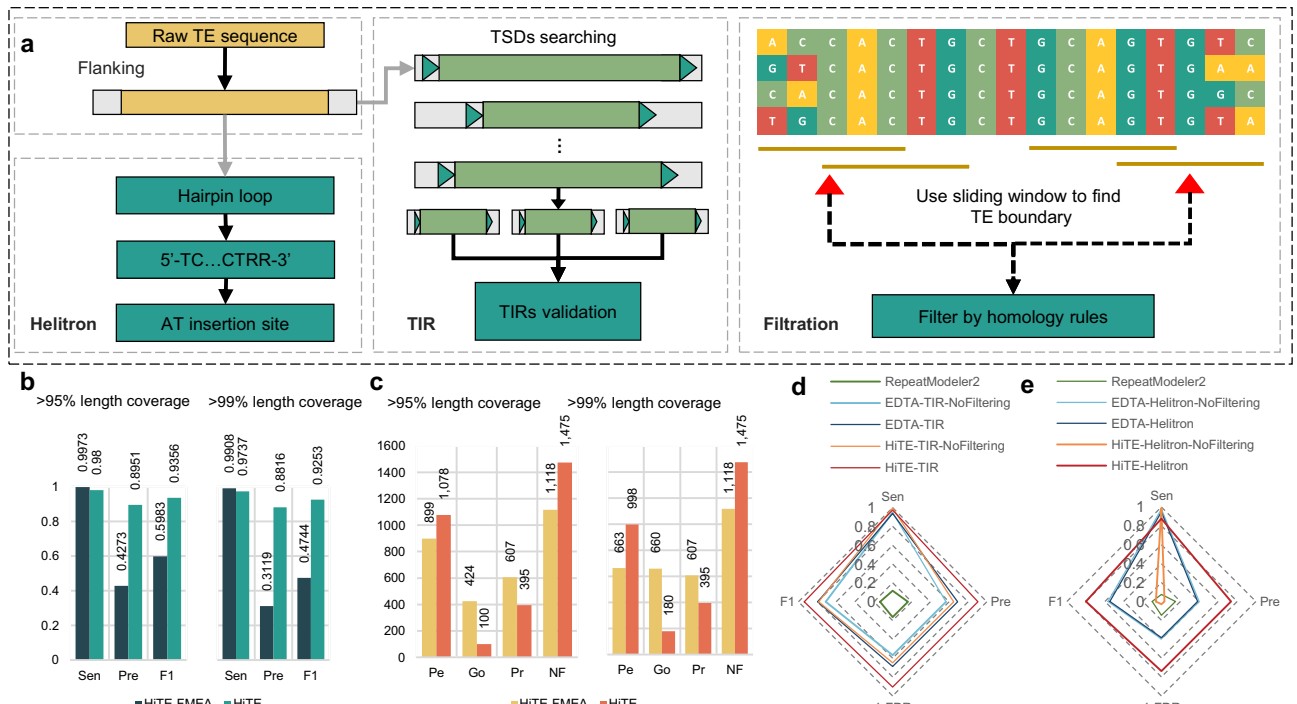

**Fig. 4 | Identification of TEs with fine-grained TE boundaries based on *O. sativa*.** **a** Pipeline of HiTE for identifying TIR and Helitron elements. Searching for confident structures reveals the true ends of TIR and Helitron elements. The robust evidence of real TEs is checked by non-homology flanking regions of multiple sequence alignment. **b, c** Evaluation of HiTE against HiTE-FMEA using BM_HiTE and BM_RM2 under rough and precise perfect definitions, respectively. **d** Evaluation of HiTE-TIR against HiTE-TIR-NoFiltering, EDTA-TIR, EDTA-TIR-NoFiltering, and RepeatModeler2 using BM_HiTE. **e** Evaluation of HiTE-Helitron against HiTE-Helitron-NoFiltering, EDTA-Helitron, EDTA-Helitron-NoFiltering, RepeatModeler2, and EAHelitron using BM_HiTE. TIR: terminal inverted repeat; TSDs: target site duplications; Pe: Perfect; Go: Good; Pr: Present; NF: Not_found. Sen: Sensitivity; Pre: Precision; FDR: False discovery rate; F1: F1-score.

to detect more precise TE families. The principles of this filtering method and the dynamic adjustment of TE boundaries are detailed in Methods and Supplementary Note 3. We evaluate the performance of the false-positive filtration module on the identification of TIR and Helitron elements based on *O. sativa*. As shown in Fig. 4d, e, when using BM_HiTE, HiTE-TIR and HiTE-Helitron show higher overall performance compared to HiTE-TIR-NoFiltering and HiTE-Helitron-NoFiltering (Supplementary Tables 7 and 8).

When using BM_EDTA, HiTE-TIR and HiTE-Helitron also demonstrate superior overall performance compared to HiTE-TIR-NoFiltering and HiTE-Helitron-NoFiltering (Supplementary Table 6). At the same time, HiTE identifies longer TE families and produces smaller but more precise library than HiTE-NoFiltering (Supplementary Fig. 7). Compared to the filtering approach of EDTA, the filtering method of HiTE proves to be more effective. For example, as shown in Supplementary Table 7, both HiTE-TIR and EDTA-TIR exhibit a similar decrease in sensitivity after using filtration methods in *O. sativa* (0.9974 to 0.9737 versus 0.9492 to 0.9385). However, HiTE-TIR shows a notable improvement in precision compared to EDTA-TIR (0.6487 to 0.9058 versus 0.5679 to 0.6897). More detailed experimental analyses regarding the filtering method are documented in Supplementary Note 4.

Reliable de novo identification of non-LTR elements is difficult due to their inherently challenging recognition characteristics. We have developed the HiTE-NonLTR module specifically designed for identifying non-LTR transposons, which searches for target site duplications (TSDs) and polyA tails for non-LTR transposons based on the outputs of our FMEA algorithm. HiTE-NonLTR exhibits excellent performance on species abundant in non-LTR elements, such as mouse and *Drosophila*. However, its performance still needs enhancement on species with limited non-LTR elements, such as Arabidopsis and rice (Supplementary Tables 9-11). Despite this, HiTE-NonLTR still

outperforms the state-of-the-art de novo non-LTR identification method, RepeatModeler2. Given that the method of HiTE-NonLTR-homology based on homologous searches exhibits superior performance on well-studied genomes, HiTE uses both HiTE-NonLTR and HiTE-NonLTR-homology to identify reliable non-LTRs.

Compared to HiTE-FMEA, which can only identify coarse-grained TE families, HiTE has achieved superior performance by implementing signature-based TE searching and filtering methods. As shown in Fig. 4b, HiTE outperforms HiTE-FMEA in both rough and precise perfect definitions. For instance, in *O. sativa*, HiTE achieves F1 scores of 0.9356 and 0.9253 under 95% and 99% length coverage thresholds, respectively, while HiTE-FMEA only attains F1 scores of 0.5983 and 0.4744. Due to the presence of numerous fragmented and false positive sequences, HiTE-FMEA generates a large library of 112 Mb. In contrast, HiTE accurately identifies TE families and filters out many fragmented sequences and false positives, resulting in a more concise library of 8.8 Mb (Supplementary Fig. 7e). At the same time, HiTE increases the number of identified *Perfect* TE families compared to HiTE-FMEA, while reducing the number of fragmented sequences represented by *Good* and *Present* TE families (Fig. 4c). As shown in Supplementary Fig. 7a, HiTE also identifies longer TE families than HiTE-FMEA. In summary, the experimental results demonstrate that the fine-grained TE boundary detection improves overall performance.

## HiTE discovers known characteristics of TEs

As an additional assessment of the ability for HiTE to discover known TEs, we run RepeatMasker[43] with each output library generated by different tools and measure the percentage of the genome masked by each major TE subclass (Fig. 5a, Supplementary Fig. 8). HiTE restores the TE landscapes of these species consistent with the curated libraries. The genome of *O. sativa* is known to contain DNA-TIR and LTR elements in close proportions[44], which is recovered by our HiTE

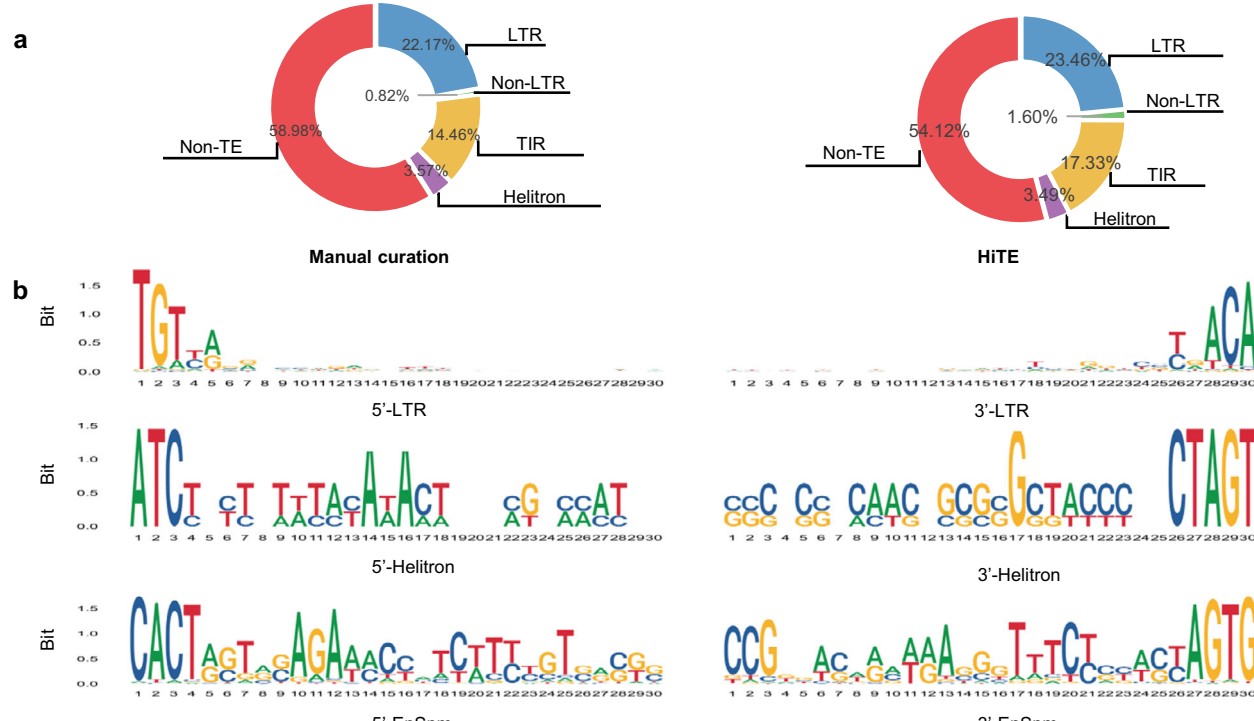

**Fig. 5 | Evaluation of HiTE on discovering the landscape and structural features of TEs. a** Percentage of the rice genome masked by each major subclass using TE libraries generated from manual curation and HiTE. **b** Sequence logos of terminal sequences of LTR, Helitron, and EnSpm elements discovered by HiTE. Only the 5′ ends and 3′ ends of the identified TEs are shown, each with a length of 30 bp. LTR: long terminal repeat; TIR: terminal inverted repeat.

library. As shown in Supplementary Fig. 8, our results show that the genome of *D. melanogaster* is dominated by retrotransposons, especially LTR and LINE retroelements[45]. The genome of *D. rerio* is dominated by class II DNA-TIR transposons, but it also has a diverse composition of LTR retroelements with many distinct families[46]. The HiTE library has discovered an abundant percentage of DNA-TIR elements in the genome of *C. briggsae*, which achieves a similar proportion to the curated library[47].

To validate the precision of the transposons identified by HiTE, we analyze the terminal sequences of LTR, Helitron, and EnSpm elements by generating sequence logos that indicate the nucleotide usage at each position in their terminals (Fig. 5b). Our findings are consistent with prior research[48], which reveals that LTR elements are generally flanked by 2-bp palindromic motifs, commonly 5′-TG...CA-3′. We have also evaluated the insertion times of two types of LTR-RTs, Copia and Gypsy (Supplementary Fig. 12), in rice and maize, and the results were consistent with previous literature[49,50]. We observe that Helitron elements are inserted into an AT target site and have no terminal inverted repeats, with the canonical terminal structure of 5′-TC...CTRR-3′ (where 5′-TC...CTAG-3′ predominates). We also note a higher AT content at the 5′ ends and an enriched CG content at the 3′ terminal, particularly at the 1st and 17th positions (Fig. 5b), which contribute to a canonical Helitron feature, such as a hairpin loop, as previously reported[51,52]. Furthermore, we identify highly conserved CACT(A/G) motifs in EnSpm short terminals, consistent with previous reports[53].

To further assess the accuracy of HiTE in TE identification, we collect eight widely recognized TIR transposons that have been extensively studied and shown to have significant impacts on genomic evolution and species diversification. These transposons, comprising six from rice, one from zebrafish, and one from maize, have been included in Supplementary Data 2. As shown in Supplementary Figs. 14–16, HiTE stands out as the sole tool among all those tested that has the capability to identify all full-length transposons. RepeatModeler2 identifies three full-length transposons, namely *Dart*, *mPing*,

and *mJing*, and fails to accurately determine the boundaries of *mPing* and *mJing*. EDTA identifies one full-length transposon, *mPing*, and shows a higher tendency to detect longer false-positive sequences that include other transposons. HiTE shows accuracy in identifying all transposons, including a unique MITE transposon called *nDart*, which is not included in other TE libraries. This transposon is absent in both the Repbase and RepeatMasker libraries, emphasizing the limitations of the current gold-standard libraries in capturing the full diversity of transposons. The experimental results further confirm the accuracy and reliability of HiTE in identifying transposons.

### HiTE discovers novel transposons
HiTE can discover novel transposons that have conservative structural features and multiple copies. TE families are usually defined using the "80-80-80" rule[34], which specifies that TE instances are members of the same family if they are longer than 80 base pairs, and share at least 80% sequence identity over 80% of their length. We cluster TIR elements output by the HiTE-TIR module and Repbase using CD-HIT[54] and the "80-80-80" rule. Clusters containing any elements from Repbase are considered known TIR elements, and the other clusters are categorized as novel. A similar method is used to distinguish novel TIR elements with new and known terminals. The novel TIR elements are realigned to the genome assembly to determine the number of their full-length copies.

We randomly select 10 novel TIR elements and obtain their copy position information on the genome. The circos plot reveals that the novel TIR elements have multiple transposition sites (Fig. 6c). To further verify their reliability, we perform a multiple sequence alignment of the copies of these novel elements using Mafft. The results clearly show the terminal inverted repeat (TIR) and target site duplication (TSD) structures, and the flanking region outside the TE structure is found to be random (Fig. 6d). The structural features of all novel TIRs discovered by HiTE are included in Supplementary Data 1, and their multiple sequence alignment files can be accessed at https://github.

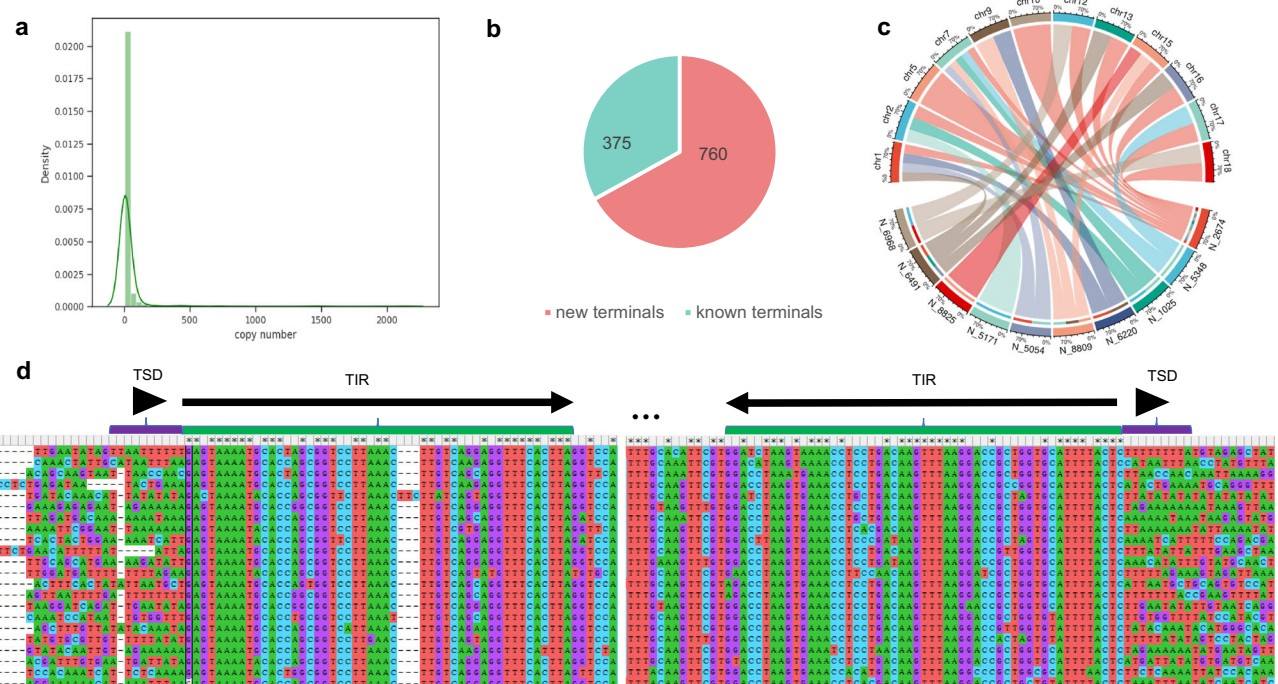

**Fig. 6 | Characteristic of novel TIR elements identified by HiTE-TIR based on *O. sativa*. a** Distribution of copy numbers of novel TIR elements. **b** The number of novel TIR elements with new and known terminals. **c** The circos plot shows that the novel TIR elements have multiple transposition sites (Only 10 randomly picked elements are shown). **d** Multiple sequence alignment of a novel TIR element. The TSD sequences and TIR sequences are designated under the black triangles and black arrows, respectively. TIR: terminal inverted repeat; TSD: target site duplication.

com/CSU-KangHu/TE_annotation. As shown in Fig. 6a, the majority of novel TIR elements are in low-copy regions with copy numbers exceeding 2, indicating that they are real TEs not detected by Repbase due to their limited number of copies. Among the novel TIR elements, 760 have new terminals, while 375 have known terminals, likely indicating that they are non-autonomous TIR elements carrying similar terminals to autonomous TIR elements[55] (Fig. 6b).

The *Ghd2* gene (*LOC_Os02g49880*) plays a crucial role in regulating key agronomic traits in rice, such as plant height, grain number, and heading date[56]. A recent study by Shen et al. has identified a stowaway-like MITE (miniature inverted-repeat transposable elements, sMITE) located in the 3'-untranslated region (3'-UTR) of *Ghd2* that functions as a translational repressor[57], which has also been confirmed by HiTE. To assess the effectiveness of HiTE in detecting transposons, we have obtained the genome assembly and annotation (genes and repeats) from the Rice Genome Annotation Project and conducted TE annotation using HiTE. The results demonstrate that the *Ghd2* gene contains a MITE (represented by the blue bar) in its UTR region, which is also identified by HiTE (represented by the green bar), as presented in Fig. 7a. The MITE is characterized as stowaway-like, based on its 'TA' TSDs and flanked by a pair of short TIRs. An interesting finding is that HiTE and RepeatModeler2 both detect a novel TIR element adjacent to the sMITE, which we have designated as *Novel_TIR_Ghd2* for the purpose of convenience (Fig. 7a). To confirm that *Novel_TIR_Ghd2* is a genuine transposon, we use the itrsearch tool, included in TE Finder 2.30 (https://github.com/urgi-anagen/TE_finder), to align the TIRs and search for the TSDs in its copies. Multiple sequence alignments of its copies are also generated using Mafft, revealing the structure of *Novel_TIR_Ghd2* (Fig. 7e). Our analysis confirms that *Novel_TIR_Ghd2* is a bona fide transposon with 9-10 bp TSDs and 224 bp TIRs, belonging to the Mutator superfamily according to Wicker et al.[34].

Given that the sMITE frequently inserts into other transposons (Fig. 7d), we hypothesize that *Ghd2* gene was originally regulated by *Novel_TIR_Ghd2* in ancient rice. However, the insertion of the sMITE into *Novel_TIR_Ghd2* resulted in the alteration of its original regulatory role, ultimately phasing it out, which may have played a pivotal role in the evolution of rice. To justify our conjecture, we have compared the copy number and TIR identity[50,58] of sMITE and *Novel_TIR_Ghd2* (Fig. 7b). The results indicate that sMITE, with 429 copies and a higher terminal identity, exhibits higher activity compared to *Novel_TIR_Ghd2*, which only has eight copies. Given the rice mutation rate of $1.3 \times 10^{-8}$ mutations per site per year, the estimated insertion times for sMITE and *Novel_TIR_Ghd2* are 5.26 and 9.48 million years ago (Mya), respectively. While EDTA appears to discover the *Novel_TIR_Ghd2* region, its identified boundaries are imprecise (Fig. 7a). In comparison, RepeatModeler2 locates the region of *Novel_TIR_Ghd2* close to HiTE, but the identified boundaries are truncated upon sequence examination, making it impossible to identify structural features such as TSD without subsequent manual editing (Fig. 7c).

Recently, Chen et al. accomplished a complete telomere-to-telomere assembly of the maize genome[59], Zm-Mo17-REFERENCE-CAU-2.0, which provides a comprehensive assembly of all 10 centromeres. This assembly enables a better understanding of centromeric transposons and their potential impacts. As shown in Supplementary Fig. 20, HiTE has discovered numerous novel TIR and Helitron transposons distributed across the centromeres and genome regions compared to the TE library from Maize TE Consortium (MTEC; https://github.com/oushujun/MTEC). Simultaneously, HiTE identifies a higher number of transposons that possess significant biological relevance, characterized by longer internal sequences and a greater abundance of protein-coding genes. In summary, HiTE can accurately identify TE instances and facilitate the discovery of new insights.

## Discussion

The rapid advancement of sequencing technologies has led to more reliable genome assemblies, which gives a bright future to comprehensive annotation of TEs. However, inaccurate TE identification tools can produce TE libraries containing many errors, which can propagate

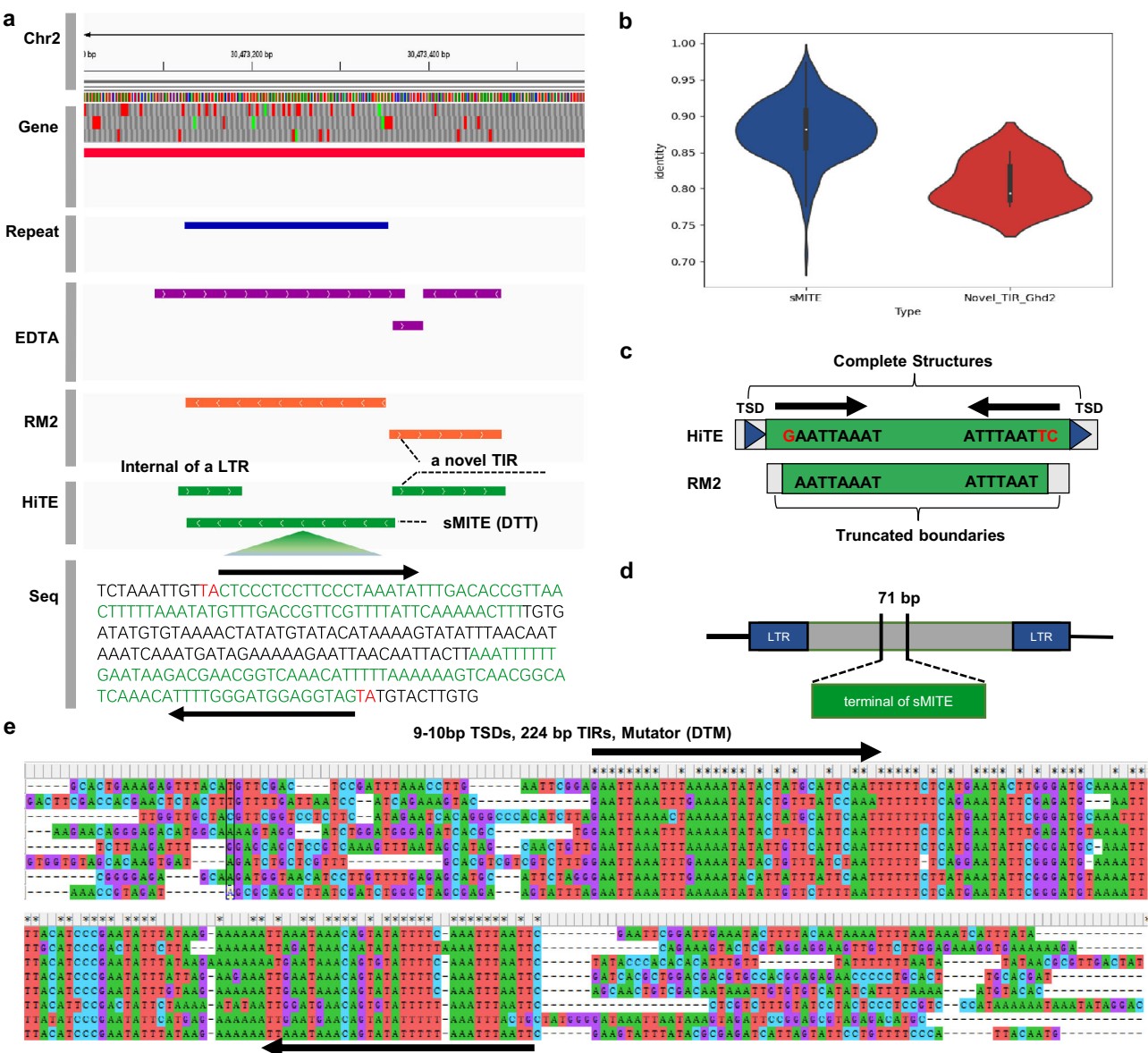

**Fig. 7 | HiTE discovers an unrevealed phenomenon between the MITE and a key gene, _Ghd2_ in rice. a** Screenshot of the Integrative Genomics Viewer (Chr2:30,473,052-30,474,405) on a sMITE inserted into the UTR region of gene _Ghd2_. Red, blue, and green bars represent the annotation of genes, repeats from RepeatMasker, and TEs from HiTE, respectively. The red and green letters in the sequence row represent the TSDs and TIRs of the sMITE, respectively. **b** Distribution of TIR identity differences in the copies of sMITE and _Novel_TIR_Ghd2_. A total of 429 copies of sMITE and 8 copies of _Novel_TIR_Ghd2_ are obtained. The boxes inside the violin plots indicate the 50th percentile (middle line), 25th, and 75th percentile (box). **c** HiTE recognizes the complete structure of _Novel_TIR_Ghd2_, while RepeatModeler2 only recognizes its incomplete boundaries. **d** One terminal of sMITE (71 bp) is discovered in the internal of an LTR-RT. **e** The complete structure of _Novel_TIR_Ghd2_ is revealed by the multiple sequence alignments of its copies. LTR: long terminal repeat; TIR: terminal inverted repeat; TSD: target site duplication; sMITE: stowaway-like miniature inverted-repeat transposable element; DTT: DNA/TIR/Tc1-Mariner; DTM: DNA/TIR/Mutator.

throughout the whole-genome annotation process[26]. In this study, we have developed and validated HiTE, an accurate dynamic boundary adjustment method for detecting intact TEs. We have developed multiple algorithms that make full use of the repetitive nature, conserved motifs, and structural features of TEs for accurate detection. We have demonstrated that HiTE can identify many novel and genuine transposons, which can serve as a valuable complement to the currently gold standard database. In addition, HiTE exhibits significantly reduced running time compared to EDTA and RepeatModeler2, greatly enhancing the efficiency of the complete genome annotation process for biologists.

Given that the protein sequence required for transposition is contained within the full-length TE, the quantity of full-length TE models is the most meaningful metric for evaluating TE integrity and biological significance. A key feature of HiTE is its capability to identify full-length TEs, even in the presence of large gaps caused by insertion, deletion, and nested TEs. At the same time, accurately determining the boundaries of TEs poses a significant challenge for automated methods, frequently requiring extensive manual identification and correction. By gradually discovering the full-length TEs from coarse- to fine-grained boundaries, HiTE can identify the precise boundaries of TEs and reduce the need for manual intervention. By benchmarking on a diverse set of species with different TE landscapes, we have demonstrated that HiTE outperforms other TE identification methods, including EarlGrey, RepeatModeler2, EDTA, and RepeatScout, in identifying intact TE insertions. Additionally, HiTE achieves superior performance in identifying specific types of TEs, such as TIR and Helitron elements. Furthermore, with the proliferation of the T2T

genome, we have demonstrated that the use of T2T assemblies in HiTE can yield favorable TE annotation results compared to assemblies with gaps (Supplementary Table 12).

Currently, methods that use genome assembly for TE identification, including HiTE, have certain limitations. Firstly, HiTE uses LTR_retriever for LTR identification, potentially resulting in the exclusion of LTRs containing extensive internal tandem repeats, like the Dasheng LTR in rice[60]. Nevertheless, HiTE can still detect LTRs lacking significant tandem repeats, even when they are inserted into centromeric or telomeric regions (Supplementary Note 10). Next, the weak structural characteristics of certain types of TEs may result in the inclusion of false-positive sequences. Although we have greatly improved the identification performance of TIR and Helitron elements, there is potential for improvement. For example, incorporating more comprehensive hairpin loop patterns in Helitron identification tools could improve their performance. Lastly, HiTE may inevitably loss some of genuine TEs. To reduce false positives, TE candidates with divergent terminals or TSDs, or accidental homology outside the boundaries, are considered false positives and discarded, which may result in the loss of some real TE instances. Detailed analysis of the unique, shared, and missing TE families in HiTE has been performed based on *Arabidopsis thaliana* (Supplementary Note 1).

In summary, through the utilization of sensitive alignment algorithms and structural feature recognition, HiTE has demonstrated considerable potential as a reliable tool for TE identification and annotation using genome assembly. We anticipate that the methods proposed in this study will contribute to the analysis of genome variation research, with potential applications in areas such as human disease and crop breeding.

## Methods

### Data preparation
During the evaluation of HiTE, nine species, namely *Oryza sativa* (assembly IRGSP-1.0), *Caenorhabditis briggsae* (assembly CB4), *Drosophila melanogaster* (assembly Release 6 plus ISO1 MT), *Danio rerio* (assembly GRCz11), *Zea mays* (assembly Zm-B73-REFERENCE-NAM-5.0), *Arabidopsis thaliana* (assembly TAIR10.1), *Gallus gallus* (assembly GCF_000002315.5), *Taeniopygia guttata* (assembly GCF_000151805.1), and *Mus musculus* (assembly GCA_000001635.2), were used as reference genomes. The curated libraries were obtained from RepBase26.05, and the appropriate parameters were used to generate TE libraries of RepeatScout, EarlGrey, RepeatModeler2, and EDTA (Supplementary Table 14). A non-LTR library was generated by extracting known LINEs and SINEs from the Dfam library of Repeat-Masker version 4.1.1 (http://www.repeatmasker.org), which is a public TE database available under the Creative Commons Zero (CC0) license. Additionally, another rice genome assembly (Oryza sativa L. ssp. japonica cv. "Nipponbare" v. MSU7)[61] was utilized to verify that HiTE can identify known and unknown transposons. The telomere-to-telomere assembly of the maize genome[59], Zm-Mo17-REFERENCE-CAU-2.0, was used to demonstrate that HiTE discovers numerous novel TIR and Helitron transposons distributed across the centromeres and genome regions compared to the TE library from Maize TE Consortium. The telomere-to-telomere assemblies of Arabidopsis[62] and rice[63] were used to assess the influence of different assembly qualities on the performance of HiTE.

### Fault-tolerant mapping expansion algorithm of HiTE (HiTE-FMEA)
The identification of a single TE instance as multiple fragments can significantly hinder the accurate identification and classification of complete TE families[64,65]. To overcome this issue, we have developed a fault-tolerant mapping expansion algorithm, termed FMEA, which can span large gaps and identify the coarse-grained boundaries of TEs. The process of FMEA involves the following steps (Fig. 3a):

(1) Sensitive pairwise alignment. We use BLASTN[66] for pairwise alignment of the genome assembly sequences. To speed up this process, assembly sequences are segmented into smaller fragments of 100 Kb using the "--chrom_seg_length" option, and a concurrent Python package is utilized for parallel acceleration[67].

(2) Fault-tolerant expansion. For each query sequence, adjacent alignments are first gathered based on their alignment positions on the target sequences, which are then sorted ascendingly for clustering. Next, within each cluster, the alignment is expanded if the next alignment is within its adjacent area, spanning alignment gaps caused by TE variations. As shown in Fig. 3a, two TE instances included in the query sequence, *a-d* and *e-f*, are identified by FMEA. Finally, each query sequence could align to multiple target sequences and generate multiple candidate instances, which are subsequently screened for redundancy. The longest sequences are taken, which usually contain one or more repeats. The algorithmic details of FMEA can be found in Supplementary Note 5.

### Structural-based TE searching of HiTE
We have developed structural-based methods for TE identification, which exploit the characteristic features of TEs, such as long terminal repeats (LTRs) and terminal inverted repeats (TIRs) at both ends of LTR and TIR elements. Additionally, TEs are typically inserted into the genome with the formation of two short TSDs resulting from the repair of DNA double-strand breaks at the integration site[68]. The sizes of the TSDs can be used as a diagnostic feature for TE identification and classification. The structural-based TE searching methods of HiTE mainly include the following parts:

(1) The identification of LTR retrotransposons (LTR-RTs). LTR-RTs are facilitated by their distinctive structural characteristics, including long direct repeat sequences ranging from 85 to 5000 bp, 2-bp palindromic motifs (5'-TG...CA-3') at both ends, and 4-6 bp TSDs flanked at the insertion site (Fig. 1h). To identify LTR-RTs, we have employed the parallel version of LTR_harvest and LTR_Finder[26,69] with default parameters (Supplementary Table 14). We then use LTR_retriever as a stringent filtering method for the candidate LTR-RTs.

(2) The identification of TIR elements. TIR elements have terminal inverted repeat sequences (usually a few bp to hundreds of bp) and conserved motif characteristics of specific superfamilies. Due to their short termini, TIR elements can be difficult to identify. To discover the structurally intact TIR elements, we first use our FMEA algorithms to find the coarse-grained boundaries of candidate TEs, which are then extended by a certain length to search for all legal TSDs (Fig. 4a). To reduce false positives, we identify TSDs that are identical. Next, we use the itrsearch tool, included in TE Finder 2.30 (https://github.com/urgi-anagen/TE_finder), to determine if the candidates possess terminal inverted repeats (Supplementary Table 14). Since a sequence with a coarse-grained boundary may produce multiple candidates with legal TIRs and TSDs, we select the candidate that is closest to the coarse-grained boundaries as the TIR-like elements. Finally, the TIR-like elements are passed to the false-positive filtering module of HiTE to search for homology boundaries, adjust the true ends of TEs, and obtain confident TIR elements.

(3) The identification of Helitron elements. Helitrons are a class of TEs that replicate through the rolling circle mechanism, and their weak structural signals make them particularly challenging to identify. To identify Helitrons, we first use EAHelitron and HelitronScanner on the candidates with coarse-grained boundaries and search for the hairpin structures (Fig. 4a). Then, we select the candidates with complete Helitron structures flanked by 5'-A and 3'-T that are closest to the coarse-grained boundaries as the Helitron-like elements. Finally, the Helitron-like elements are passed to the false-positive filtering module of HiTE to search for homology boundaries, adjust the true ends of TEs, and obtain confident Helitrons.

(4) The identification of non-LTR elements. The identification of non-LTR elements, such as LINEs and SINEs, is challenging due to their variability and lack of discernible structural signals[70,71]. The most effective method for non-LTR identification involves searching known non-LTR libraries and relying on extensive expert knowledge for manual curation. To equip HiTE with the ability to identify non-LTR elements, we have developed a de novo module for non-LTR detection, namely HiTE-NonLTR. It begins by using the coarse-grained repeats generated by our FMEA algorithm. Subsequently, it searches for structural characteristics such as TSDs (8-20 bp) and polyA tails (exceeding 6 bp). Candidate non-LTR elements are then aligned to the genome to obtain their full-length copies, followed by multiple sequence alignment of all copies. Next, we have designed a dynamic boundary adjustment method to search for the homologous boundaries of the copies, determining the raw 5′ and 3′ ends. Each copy is then examined to identify the polyA tail near the raw 3′ end, establishing the true 3′ end of each copy. With the 3′ end established, all possible 3′ end TSDs are obtained, and corresponding 5′ end TSDs are searched near the raw 5′ end to determine the true 5′ end of each copy. To minimize false positives, we require that a genuine non-LTR element must have a copy count with TSDs greater than half of all copy counts, or exceed 5. While this stringent filtering method ensures reliability in identification, it may result in the loss of some intact LINE elements with low copy numbers. Considering the conservative nature of domains across various TEs, we use a curated LINE domain library to retain candidate LINE elements harboring intact domains. The curated LINE domain library is derived from the RepeatPeps.lib within RepeatMasker, which is also used in the classification module of RepeatModeler2. HiTE-NonLTR exhibits excellent performance on species abundant in non-LTR elements, such as mouse and Drosophila. However, its performance still needs enhancement on species with limited non-LTR elements, such as Arabidopsis and rice (Supplementary Tables 9-11). To address this issue, we have also developed a high-precision annotation approach called HiTE-NonLTR-homology, specifically tailored for non-LTR elements. This approach involves the creation of a non-LTR library, incorporating known LINEs and SINEs from the Dfam library of RepeatMasker version 4.1.1. Subsequently, this library is used to search against the genome to identify confident non-LTR families. By combining de novo and homology-based methods, HiTE achieves reliable non-LTR identification across different species.

## False-positive filtering of HiTE

False positives in the library can propagate throughout the whole-genome annotation processes. The false positive filtering methods of HiTE are summarized in Supplementary Fig. 4. We describe three major types of false positives that significantly affect the accuracy of TE identification:

(1) Tandem repeat. We use Tandem Repeats Finder (TRF) to mask tandem repeats with parameters "2 7 7 80 10 50 500 -f -d -m" (Supplementary Table 14). Furthermore, TIR candidates with tandem repeats exceeding 10 bp at the beginning of their terminals are also eliminated to avoid fake TIR elements, such as those that begin with 'TA' and end with 'AT', which are frequently associated with simple tandem repeats.

(2) Spurious TIR elements with LTR terminals. The occurrence of short TIR terminals with legal TSDs within the LTR-RTs may generate spurious TIR candidates, leading to false positive results. To mitigate such outcomes, TIR candidates are aligned to the LTR-RTs identified by the LTR searching module of HiTE. TIR candidates with more than 95% overlap regions with LTR-RTs are regarded as false positives and subsequently removed from further analysis.

(3) Candidates with homologs beyond their copy boundaries. Many false positive sequences may possess TE-like structures by chance, leading to a significant amount of misclassification. To address

this issue, we have developed a novel homology-based filtering method based on two principles, as shown in Fig. 4a. First, transposons should occur at least twice in the genome, disregarding old TEs that have evolved and diverged significantly over time. Second, the region outside the boundaries of TEs should be composed of random sequences without homologs. Therefore, we first extend both ends of the candidate TE copies and perform multiple sequence alignments between them. Subsequently, we use a 10-bp sliding window to detect homology boundaries in the multiple sequence alignment files. We then evaluate whether the candidate transposon identified by the homologous boundary met a specific transposon structure, such as the presence of target site duplications (TSDs) and terminal inverted repeats for TIR transposons and 5′-ATC…CTRR-3′ structures for Helitron transposons. Ultimately, we obtain a set of genuine TEs with well-defined structures. As shown in Supplementary Note 3, we use an example to show how we can go from candidates with coarse-grained boundaries to gradually search for homology boundaries and obtain confident TIR elements.

## Unwrapping nested TEs

Nested TEs, which are transposons inserted into other transposons, can complicate the identification of individual TE sequences. To address this issue, HiTE uses a three-step process to unwrap the nested TEs. Firstly, HiTE removes the full-length TEs contained in other sequences with more than 95% coverage and 95% identity and connects the remaining sequences. Secondly, sequences shorter than 100 bp are filtered out, while the remaining sequences are treated as new TE sequences. Finally, the process is iterated several times to unwrap heavily nested TEs. This process allows for the accurate identification and annotation of individual TE sequences even in the presence of nested TEs.

## Generation of a classified TE library

HiTE uses different consensus calling methods for various types of TEs. For LTR transposons, we first conduct all-vs-all BLASTN alignments based on the results obtained from LTR_retriever. Then, we apply our FMEA algorithm to bridge the gaps caused by large deletions, allowing us to cluster sequences spanning these gaps. Following this, we perform multiple sequence alignment using MAFFT and generate a consensus sequence based on the majority rule. For TIR, Helitron, and non-LTR elements, we use MAFFT based on the majority rule for consensus calling. We employ CD-HIT-EST with the parameter "-aS 0.95 -aL 0.95 -c 0.8 -G 0 -g 1 -A 80" for clustering all types of TEs. Since most TE instances identified by HiTE are full-length, we opt for a more stringent rule of 95-95-80-80, as opposed to the conventional 80-80-80 rule. Before clustering, LTR-RTs are divided into 5′ LTRs, 3′ LTRs, and LTR internal regions. HiTE offers two approaches for TE classification: RepeatClassifier and NeuralTE. RepeatClassifier employs homology-based searches to classify TEs with sequence similarity to known transposons. On the other hand, NeuralTE uses a deep learning method to achieve superior classification for novel transposons by identifying various structural features of transposons, such as TSDs, 5-bp ends, domains, terminal, and internal information[39].

Many TEs contain open reading frames (ORFs) that encode proteins necessary for their transposition. Predicting conserved protein domains in TEs can help with their classification into the different superfamilies. We perform the homology search between the protein database and TE models using BLASTX and filtered out bad hits with e-value cutoff 1e-20. The protein database of known TE peptides can be downloaded from RepeatMasker. In addition, alignment results are often fragmented, posing difficulty in discovering the intact protein domains of TEs. To address this issue, we connect and bridge the fragmented alignments to create a more continuous domain area within the TE families. Finally, we generate a table that maps TE families

to the corresponding protein domain locations, providing a convenient description of the protein structure of TEs.

## Evaluation method of HiTE

To evaluate the performance of HiTE and compare it with other TE detection tools, three benchmarking methods are employed.

(1) Evaluation by the benchmarking method of RepeatModeler2 (BM_RM2). BM_RM2 performs an alignment of the tested TE library with a gold standard library and categorizes the resulting matches into four levels: "*Perfect*", "*Good*", "*Present*", and "*Not found*". Please refer to Supplementary Note 2 for more specific details.

(2) Evaluation by the benchmarking method of EDTA (BM_EDTA). BM_EDTA evaluates the performance of various tools by annotating the genome with the gold standard TE library and the tested TE library generated by different tools. The assessment is conducted based on a summary of the total number of genomic DNA bases. Six metrics comprising sensitivity, specificity, accuracy, precision, FDR (False Discovery Rate), and F1, are used to characterize the annotation performance of the tested library.

(3) Evaluation by the benchmarking method of HiTE (BM_HiTE). To better assess the quality of the test library, we have designed a new evaluation approach, referred to as BM_HiTE. As shown in Supplementary Fig. 1, the major difference between BM_HiTE and BM_EDTA lies in the criteria for calculating true positives. BM_EDTA considers any sequence that has any length of match with the standard library as a true positive. In BM_HiTE, a sequence is considered a true positive only when the length of the overlap between the test and Repbase TE sequence exceeds a threshold, such as 80% of their lengths respectively. All others are considered as false positives, including alignments with significant shifts, longer sequences containing the true TE, and fragmented TE sequences. BM_HiTE mitigates the influence of fragmented sequences on the results, making it more suitable for evaluating high-quality full-length TE libraries.

## Reporting summary

Further information on research design is available in the Nature Portfolio Reporting Summary linked to this article.

## Data availability

The reference genomes for nine species, including *Oryza sativa* (assembly IRGSP-1.0), *Caenorhabditis briggsae* (assembly CB4), *Drosophila melanogaster* (assembly Release 6 plus ISO1 MT), *Danio rerio* (assembly GRCz11), *Zea mays* (assembly Zm-B73-REFERENCE-NAM-5.0), *Arabidopsis thaliana* (assembly TAIR10.1), *Gallus gallus* (assembly GCF_000002315.5), *Taeniopygia guttata* (assembly GCF_000151805.1), and *Mus musculus* (assembly GCA_000001635.2), can be accessed through NCBI GenBank [https://www.ncbi.nlm.nih.gov/genome/]. The other rice genome (Oryza sativa L. ssp. japonica cv. "*Nipponbare*" v. MSU7) used in the *Ghd2* gene experiment of this study, as well as its annotation with respect to both genes and repeats, can be accessed through the Rice Genome Annotation Project [http://rice.uga.edu/]. The telomere-to-telomere assembly of the maize, rice, and Arabidopsis genomes used in this study can be found in CyVerse [https://data.cyverse.org/dav-anon/iplant/home/laijs/Zm-Mo17-REFERENCE-CAU-2.0/], RiceSuperPIRdb [http://www.ricesuperpir.com/web/download], and GitHub [https://github.com/schatzlab/Col-CEN/tree/main/v1.2]. The curated TE libraries used in this study can be accessed through a paid subscription to Repbase [https://www.girinst.org/repbase/]. Additionally, the TE libraries and novel transposons generated in this study are publicly available in the GitHub repository CSU-KangHu/TE_annotation [https://github.com/CSU-KangHu/TE_annotation] and Zenodo[72].

## Code availability

HiTE is publicly available at GitHub [https://github.com/CSU-KangHu/HiTE] and Zenodo[73].

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

## Acknowledgements

This work was supported in part by the National Key Research and Development Program of China (No.2021YFF1201200), the National Natural Science Foundation of China under Grants (Nos. 62350004, 62332020), the Science Foundation for Distinguished Young Scholars of Hunan Province (NO. 2023JJ10080), Fundamental Research Funds for the Central Universities of Central South University (2021zzts0208). This publication is partially based upon work supported by the King Abdullah University of Science and Technology (KAUST) Office of Research Administration (ORA) under Award No REI/1/5234-01-01, REI/1/5414-01-01, and REI/1/5289-01-01. This work was carried out in part using computing resources at the High Performance Computing Center of Central South University.

## Author contributions

J.X.W. and B.H. conceived and designed this project. K.H., P.N., J.X.W. and B.H. conceived, designed, and implemented the HiTE. J.R. and J.Y.C. provided valuable guidance drawing on their expertise in biology. X.G. helped download the Repbase database. K.H., P.N. and M.H.X. conducted the analyses. Y.Z. helped evaluate HiTE on the HPC cluster. K.H., J.X.W. and B.H. wrote the paper. X.G. and Y.H.L. revised and proofread the manuscript. All authors have read and approved the final version of this paper.

## Competing interests

The authors declare no competing interests.
