## [Peer Review File · Nature Communications]

HiTE: A fast and accurate dynamic boundary adjustment approach for full-length Transposable Elements annotationREVIEWER COMMENTS

Reviewer #1 (Remarks to the Author):

There are many existing TE annotation pipelines, and this manuscript describes the development of another. The novel portion of the paper highlights the use of a sliding window approach to determine TE boundaries, and important step for TE annotation. However, the authors do not accurately describe the differences between their pipeline and others, which only serves to add confusion to an already complex field. The authors also promise the development of tools, but other than the boundary-finding module, there is merely a filtering of other, previously-published tools. Even the assessment tools and methods are not novel.

Title

Please put "HiTE" in the title. Difficult to find pipeline papers without the name of the pipeline in the title.

Results

"HiTE is an automated TE annotation pipeline that combines the strengths of de novo and signature-based methods, aiming to produce a high-quality TE library."

-This is the same goal as RepeatModeler2 and EDTA. How is yours different?

"HiTE uses multiple filtering methods to ensure that the candidates discovered have complete TE structures"

EDTA and RepeatModeler2 use lots of filtering as well. Highlight the differences.

"As shown in Supplementary Table 2, we notice that HiTE exhibits relatively lower sensitivity. To explore further, we conduct a manual analysis of the missed TIR transposons in *Arabidopsis thaliana*." Supplementary Table 2 is large with a lot of numbers and acronyms. It's difficult to parse out exactly what the authors are referring to when they state that there is "relatively lower sensitivity". Also, what does this mean? Is this compared to other pipelines?

Authors spend a page or two explaining how their pipeline works and the different modules. These same modules can be found in other pipelines as well. What makes this program perform better?

Do the authors have the necessary rights to use the Repbase library?

" Therefore, we can only evaluate HiTE and EDTA separately based on their recognition modules for different types of transposons."

RepeatModeler2 contains an optional LTR detection parameter.

"It is important to note that although the results of RepeatModeler2 may appear to be categorized, we cannot directly extract different types of transposons based on the classification labels, as the classification module relies on known TE databases to annotate the identification results, thereby introducing prior information."

Earlier in the text it was stated that the classification module of RepeatModeler2 was utilized for classification purposes in this pipeline. If it's not accurate, based on the reasons you indicate, why have you chosen to use it for your pipeline?

" Moreover, HiTE-Helitron identifies longer TE models and generates a considerably smaller library in comparison to HiTE-Helitron-NoFiltering."

Is this a reduction in redundancy?

" We developed structural-based methods for TE identification, which exploit the characteristic

features of TEs, such....”

While this pipeline filters the data, there is no development of unique tools to handle various TE types.

Methods

Questions:

1. What are the sizes of the “genome chunks” during first step?
2. I cannot properly access your code. Please send other file format.
3. What is the output type of your program? E.g., fasta sequences, bed files, GFF, etc.
4. Where/what are the intermediate files? This is to ensure that the provenance of the models is maintained and fosters reproducibility as well as understanding the data.
5. Which consensus caller does the pipeline use? This can greatly affect the results.

Your RepeatMasker settings:

```
RepeatMasker -lib repbase_lib -nolow -pa threads test_lib
```

will miss older families, as the search stages associated with the -lib option are greatly reduced as compared to the -species option. In addition, adding the -nolow option will increase the number of false positives in your output.

Neither the main nor supplementary text contains any references.

Reviewer #2 (Remarks to the Author):

Hu et al. present the development of HiTE for fast and accurate annotation of full-length TEs in both plant and animal genomes. I find this topic of high importance since sequencing and assembly of eukaryotic genomes have been much improved, yet genome annotation is still lagging with slow speed and inaccurate results. The proposed approach is very promising, showing much-improved precision and full-length TE detection. However, the sensitivity seems to be over-sacrificed to achieve high precision, resulting in an overall lower sensitivity than companion tools. TIR detection of HiTE is very promising, yet with the lack of further classification into clades. The execution speed of the program still has room to improve to claim as “fast.” Overall, I think HiTE is very promising but may need further improvement to become a useful tool for the genome annotation community.

Major

1. RepeatModeler2 has pretty consistent performance in terms of execution time, which could benefit large genome annotation. For example, RM2 spent 24 hours in maize, while HiTE took over 32 hours. The reviewer tested on slightly larger genomes, such as human and maize, the HiTE ran very slowly compared to other programs. The author claims this is a fast program in their title but testing on the human genome costed almost 10 days using 20 CPU threads, which is not as fast as I thought it should be.

2. Suppl. Table 2. Why is HiTE-FEMA only available in rice? It seems to me it’s still an experimental module that is not readily available on other species.

3. Although it is used in the RepeatModeler benchmarking, it is still inaccurate and misleading to call matches with >95% similarity and >95% coverage “Perfect,” which is defined as “having all the required or desirable elements, qualities, or characteristics; as good as it is possible to be.” Many structural tools search for perfect intact TEs to make high-quality libraries, such as LTR_retriever, but the definition used in this study is rough, and applying such metric on tools that are more stringent on the criteria or use a totally different approach to identify repeats would result in misleading results. For example, RepeatModeler2 uses an all-by-all blast approach to identify repeat units and is not designed to identify “perfect” intact TEs (Fig. 2a).

4. "However, the evaluation of BM_EDTA relies on the count of matched bases instead of the complete TE sequence, potentially leading to the inclusion of short false-positive sequences or fragments, which in turn can cause an erroneous elevation of sensitivity." As far as I know, EDTA has two approaches to annotating TEs. The first approach uses structural-based methods to identify full-length TEs (the "Perfect" category). The second approach uses RepeatMasker for homology annotation of fragmented TEs (The Good and Present categories). The authors refer to the homology annotation approach, which effectively suggests that RepeatMasker could include short false-positive sequences. Such a claim is not supported by their data nor by other publications, which can be viewed as an excuse for their low sensitivity results. In fact, the authors seem to be biased toward the complete TE sequence, which consists of only a small fraction of most eukaryotic genomes. Most genomes are full of fragmented TE sequences because of genome purging.

5. In rice, the non-filtered Helitorns identified by HiTE have pretty high sensitivity (90%), but when filtering is applied by HiTE, it has low sensitivity (61.3%), suggesting the filtering criteria are not very specific. In Arabidopsis, the trend is similar, except the non-filtered Helitorns by HiTE have quite a low sensitivity (70.7%), resulting in a further lower sensitivity when filtered by HiTE (55.4%). See Fig 4e. Again, the sensitivity performance is suboptimal in HiTE, which is critical for genome annotations.

6. Non-LTR identification relies on existing libraries of the species to be annotated, which is a limiting factor in the application of HiTE on non-model species without existing non-LTR libraries. In species without manually curated SINE/LINE libraries, the annotation sensitivity of HiTE will be very low, if not 0.

7. Classification failed on all genomes tested on the reviewer's end. Furthermore, the reviewer used the rice result to benchmark with the EDTA lib-test.pl script, and obtained only 57.7% sensitivity on the total category (all repeats due to the lack of classification).

8. Precision is calculated as the proportion of true positives on all positive results, which completely ignores false negatives (suppl. Fig 1a). You can have a very high precision, meaning most of the reported TEs are true TEs, and also a very high false negative, meaning low sensitivity. The authors emphasize on high precision performance of HiTE but not so much for sensitivity because, except for TIR, other TEs have low sensitivity despite high precision.

9. TIR is a superfamily of DNA transposons that contains many clades such as PILE, CMC-EnSpm, hAT, MuDR, piggyBac, TcMar_Stowaway, etc. The authors did not demonstrate HiTE's capability of classifying these clades, which is a critical defect as a comprehensive TE annotation tool.

Minor:

Supp Table 1, Total is located in the middle of other TE superfamilies, which should be the last entry of each species.

Suppl. Tables and benchmark results for different species/categories should be separated by a line for better readability.

Species names should be italicized consistently.

Summary

We appreciate the valuable comments and suggestions from the editor and reviewers. Based on the suggestions and comments from editor and reviewers, we revised our paper. We addressed those comments and suggestions carefully and included a point-by-point response below. The significant changes in the revised manuscript were highlighted by red color.

Answers to Reviewer #1

Reviewer #1 (Remarks to the Author):

Comment 1. *There are many existing TE annotation pipelines, and this manuscript describes the development of another. The novel portion of the paper highlights the use of a sliding window approach to determine TE boundaries, and important step for TE annotation. However, the authors do not accurately describe the differences between their pipeline and others, which only serves to add confusion to an already complex field. The authors also promise the development of tools, but other than the boundary-finding module, there is merely a filtering of other, previously-published tools.*

Authors' Response. Thanks for the above concern and suggestion. The *de novo* method, signature-based method, and filtering method are three common approaches in the current TE identification/annotation pipeline, used by tools like HiTE, RepeatModeler2, EDTA, and EarlGrey [1]. However, HiTE distinguishes itself from existing pipelines in the design and implementation of these three methods:

The *de novo* method: RepeatModeler2 employs RECON [2] and RepeatScout [3] for the identification of TE families. While EDTA predominantly uses the signature-based method for identifying LTR, TIR, and Helitron transposons, it employs the sequence repetition feature of RepeatModeler2 to detect non-LTR and unclassified transposons that may be missed by signature-based methods. EarlGrey employs a “*BLAST, Extract, Extend (BEE)*” strategy based on the results from RepeatModeler2, with the goal of generating maximum-length *de novo* TE consensus sequences. In contrast to the existing pipeline, HiTE develops a novel fault-tolerant mapping expansion algorithm, denoted as FMEA, to identify the full-length TEs.

As shown in Supplementary Fig. 3(A), methods such as RECON have been used to cluster alignments, potentially leading to the creation of fragmented TE families due to the excessive fragmentation in alignments. In contrast, our FMEA algorithm is not primarily concerned with identifying precise TE families. Instead, it is designed to connect and span insertions and gaps between alignment segments, aiming to preserve the integrity of TE instances as much as possible (Supplementary Note 5).

The FMEA algorithm is the *de novo* TE searching module of HiTE, which provides the input for the subsequent signature-based TE searching module. As shown in Supplementary Table 2, the FMEA algorithm outperforms other pipelines in identifying more full-length TE families.

The signature-based method: We elucidate the differences between HiTE and existing pipelines in the identification of four types of transposons: LTR, TIR, Helitron, and non-LTR. To our knowledge, neither RepeatModeler2 nor EarlGrey has designed signature-based modules specifically for TIR, Helitron, and non-LTR transposons.

(i) For the identification of LTR transposons, HiTE, EDTA, and RepeatModeler2 use the LTR_retriever [4], whereas EarlGrey relies on LTR_Finder [5] and RepeatCrat [6]. EDTA retains the raw results from LTR_retriever. In contrast, both HiTE and RepeatModeler2 develop distinct methods to reduce redundancy, a

necessity due to the frequent deletions within LTR transposons. RepeatModeler2 uses MAFFT [7] and Ninja [8] to treat gaps as a single difference, then uses Refiner [9] to generate consensus sequences. On the other hand, HiTE begins with BLASTN [10] for all-vs-all comparisons, applies our FMEA algorithm to bridge gaps, clusters sequences that bridge gaps, performs multiple sequence alignment using MAFFT, and ultimately leads to the generation of consensus sequences based on the majority rule.

(ii) For the identification of TIR and Helitron transposons, both HiTE and EDTA have developed separate modules to detect these two types of transposons. EDTA employs TIR-learner [11] for identifying TIR transposons and utilizes HelitronScanner [12] for identifying Helitron transposons. In contrast, HiTE has developed the HiTE-TIR and HiTE-Helitron modules tailored for these two types of transposons. Unlike EDTA, which uses the whole genome as input, HiTE performs identification on all potential TE candidate fragments generated by our FMEA algorithm. As shown in Supplementary Fig. 3(B), given the misleading nature of TE structural signals, the signature-based methods using the whole genome as input often lead to more false positives.

As shown in Supplementary Table 5, under the same conditions without the use of filtering algorithms, the signature-based method of HiTE (HiTE-NoFiltering) can identify more perfect TE families compared to the signature-based method of EDTA (EDTA-NoFiltering) when using the benchmarking method of RepeatModeler2 (BM_RM2). When using the benchmarking method of EDTA (BM_EDTA), HiTE-NoFiltering shows comparable or higher performance compared to EDTA-NoFiltering (Supplementary Table 6). However, since BM_EDTA calculates all bases with overlaps against the gold standard annotations, it may not be the most suitable method for evaluating high-quality full-length TE libraries. To better assess the quality of the test library, we have designed a new benchmarking method, BM_HiTE (Supplementary Fig. 2). As shown in Supplementary Table 7, HiTE-NoFiltering exhibits higher sensitivity and precision compared to EDTA-NoFiltering when evaluated using BM_HiTE. A comprehensive comparison between BM_HiTE and BM_EDTA can be found in the response to Comment 2 and Supplementary Note 2.

(iii) For the identification of non-LTR transposons, reliable *de novo* identification of non-LTR elements is difficult due to their inherently challenging recognition characteristics. EDTA does not incorporate a signature-based module for identifying non-LTR transposons. Instead, it uses the sequence repetition feature of RepeatModeler2 for non-LTR transposon identification. We have developed the HiTE-NonLTR module specifically designed for identifying non-LTR transposons, which searches for target site duplications (TSDs) and polyA/T tails for non-LTR transposons based on the outputs of our FMEA algorithm.

When using BM_EDTA, RepeatModeler2 exhibits relatively high performance on rice (Supplementary Table 10). However, we observed that RepeatModeler2 identifies only one perfect TE family, whereas HiTE-NonLTR identifies 18 perfect non-LTR families (Supplementary Table 9). As shown in Supplementary Table 11, HiTE-NonLTR shows higher sensitivity and precision compared to RepeatModeler2 when using BM_HiTE. However, the performance of *de novo* non-LTR identification remains less satisfactory. Due to the variability and lack of discernible structural signals of non-LTR elements, we have developed a homology-based non-LTR searching module called HiTE-NonLTR-homology to achieve high-precision non-LTR annotation (Supplementary Tables 9-11).

The filtering method: RepeatModeler2 uses LTR_retriever to identify LTR transposons, which play a pivotal role as the primary reference during the integration of results from RECON and RepeatScout. Furthermore, RepeatModeler2 makes use of the parameter '*-LTRMaxSeqLen*' to set a limit on the maximum length of LTR internal sequences. Additionally, RepeatModeler2 employs TRF [13] to mask tandem repeats. EarlGrey employs RepeatMasker with a conservative score threshold of 400 ('*-cutoff 400*') to eliminate weak matches that are less likely to represent genuine TE sequences. Additionally, any TE annotations shorter than 100 base pairs

are discarded. As shown in Supplementary Figs. 4 and 5, both EDTA and HiTE use various filtering methods. Among these methods, a flanking sequence-based filtering method employed by EDTA, LTR_retriever, and HiTE is particularly crucial (Supplementary Fig. 6).

As shown in Supplementary Fig. 3(C), the flanking sequence-based filtering method of HiTE differs from those of EDTA and LTR_retriever in that we have developed a novel homology-based filtering method based on multiple sequence alignment. This method not only filters out many false-positive sequences but also dynamically identifies genuine TE boundaries. As shown in Supplementary Tables 5-7, the filtering method of HiTE shows a substantial superiority over that of EDTA. The filtering method is based on the following concept: although it exhibits a bias, it is generally assumed that transposons are inserted randomly into the genome. Therefore, collecting all full-length copies of a transposon and subjecting them to a multiple sequence alignment reveals that outside the transposon boundaries, the alignment region displays randomness. In contrast, within the transposon boundaries, the alignment region exhibits a high degree of consistency since all are copies of the same transposon.

In summary, we modularized our method, where each module can be used independently as a specific TE identification tool. We have added this analysis in detail in Supplementary Note 6, Supplementary Fig. 3, and the '*Overview of HiTE*' section in the revised manuscript.

References

- [1] Baril, T., Imrie, R. M. & Hayward, A. Earl Grey: a fully automated user-friendly transposable element annotation and analysis pipeline. *bioRxiv*, 2022.2006.2030.498289, doi:10.1101/2022.06.30.498289 (2022).
- [2] Bao, Z. & Eddy, S. R. Automated de novo identification of repeat sequence families in sequenced genomes. *Genome Res.* 12, 1269-1276 (2002).
- [3] Price, A. L., Jones, N. C. & Pevzner, P. A. De novo identification of repeat families in large genomes. *Bioinformatics* 21, i351-i358 (2005).
- [4] Ou, S. & Jiang, N. LTR_retriever: a highly accurate and sensitive program for identification of long terminal repeat retrotransposons. *Plant physiology* 176, 1410-1422 (2018).
- [5] Xu, Z. & Wang, H. LTR_FINDER: an efficient tool for the prediction of full-length LTR retrotransposons. *Nucleic Acids Res.* 35, W265-W268 (2007).
- [6] Wong, W. Y. & Simakov, O. RepeatCraft: a meta-pipeline for repetitive element de-fragmentation and annotation. *Bioinformatics* 35, 1051-1052 (2019).
- [7] Katoh, K. & Standley, D. M. MAFFT multiple sequence alignment software version 7: improvements in performance and usability. *Mol. Biol. Evol.* 30, 772-780 (2013).
- [8] Wheeler, T. J. in *Algorithms in Bioinformatics: 9th International Workshop, WABI 2009, Philadelphia, PA, USA, September 12-13, 2009. Proceedings* 9. 375-389 (Springer).
- [9] Hubley, R., Wheeler, T. J. & Smit, A. F. Accuracy of multiple sequence alignment methods in the reconstruction of transposable element families. *NAR genomics and bioinformatics* 4, lqac040 (2022).
- [10] BlastN, G. BLAST: basic local alignment search tool. *NUTRITIONAL AND PHYSIOLOGICAL DISORDERS IN HORTICULTURAL CROPS* (2019).
- [11] Su, W., Gu, X. & Peterson, T. TIR-Learner, a new ensemble method for TIR transposable element annotation, provides evidence for abundant new transposable elements in the maize genome. *Molecular plant* 12, 447-460 (2019).

- [12] Xiong, W., He, L., Lai, J., Dooner, H. K. & Du, C. HelitronScanner uncovers a large overlooked cache of Helitron transposons in many plant genomes. *Proceedings of the National Academy of Sciences* 111, 10263-10268 (2014).
- [13] Benson, G. Tandem repeats finder: a program to analyze DNA sequences. *Nucleic Acids Res.* 27, 573-580 (1999).

Comment 2. *Even the assessment tools and methods are not novel.*

Authors' Response. Thank you for bringing this up. In the revised manuscript, we have introduced a new pipeline, EarlGrey [1], for comparative analysis. Another tool, tephra (<https://github.com/sestaton/tephra>), has been successfully run on the smaller genome of *C. briggsae* but faced issues with all other genomes (<https://github.com/sestaton/tephra/issues/51>). Therefore, we do not include the tool tephra for comparison.

As shown in Supplementary Fig. 1, we have employed two benchmarking methods for evaluation, namely BM_RM2 and BM_EDTA. However, BM_RM2 does not present the false positive rates in the tested TE library. BM_EDTA cannot exclude the influence of fragmented sequences, potentially leading to misleadingly high sensitivity (Supplementary Note 2). To better assess the quality of the test library, we have designed a new evaluation approach, referred to as BM_HiTE (Supplementary Fig. 2). An ideal TE library should exclusively consist of intact TE families, without any fragments [2]. However, the genome contains many fragmented copies of full-length TEs, and their proportion even surpasses that of intact full-length TEs. The fragmented TE sequences in the test library can be aligned to these fragmented copies of full-length TEs, resulting in misleadingly high sensitivity. Therefore, to accurately assess the quality of the test library, we filter out the fragmented annotation and retain only the intact full-length TE copies.

The major difference between BM_HiTE and BM_EDTA lies in the criteria for calculating true positives. BM_EDTA considers any sequence that has any length of match with the standard library as a true positive. In BM_HiTE, a sequence is considered a true positive only when the length of the overlap between the test and Repbase TE sequence exceeds 95% of their lengths respectively. As shown in Supplementary Fig. 2, all others are considered false positives, including alignments with significant shifts, longer sequences containing the true TE, and fragmented TE sequences. Compared to the benchmarking method of EDTA, BM_HiTE mitigates the influence of fragmented sequences on the results, making it more suitable for evaluating high-quality full-length TE libraries.

We have added this analysis in detail in Supplementary Note 2, Supplementary Fig. 2, and the section '*HiTE accurately detects more intact TE models*' of the revised manuscript.

References

- [1] Tobias Baril, Ryan M. Imrie, Alex Hayward et al. Earl Grey: a fully automated user-friendly transposable element annotation and analysis pipeline, 01 July 2022, PREPRINT (Version 1) available at Research Square [<https://doi.org/10.21203/rs.3.rs-1812599/v1>]
- [2] Storer JM, Hubley R, Rosen J, Smit AF. Methodologies for the de novo discovery of transposable element families. *Genes.* 2022 Apr 17;13(4):709.

Title

Comment 3.

Please put “HiTE” in the title. Difficult to find pipeline papers without the name of the pipeline in the title.

Authors' Response. Thank you for pointing this out. We have included "HiTE" in the title as suggested.

Results

Comment 4.

“HiTE is an automated TE annotation pipeline that combines the strengths of de novo and signature-based methods, aiming to produce a high-quality TE library.”

-This is the same goal as RepeatModeler2 and EDTA. How is yours different?

Authors' Response. We thank the reviewer for this question. While both HiTE, RepeatModeler2, and EDTA share the goal of generating high-quality full-length TE libraries, there are differences in their design principles and implementations:

(1) **The *de novo* method:** RepeatModeler2 employs RECON and RepeatScout for the identification of TE families. While EDTA primarily relies on the signature-based method for identifying LTR, TIR, and Helitron transposons, it uses the sequence repetition feature of RepeatModeler2 to detect non-LTR and unclassified transposons that may be missed by signature-based methods. In contrast, HiTE develops a novel fault-tolerant mapping expansion algorithm, denoted as FMEA, to identify the full-length TEs. The FMEA algorithm is designed to bridge gaps and traverse insertions between alignment segments, thereby preserving the integrity of TE structures. As shown in Supplementary Table 2, the FMEA algorithm outperforms RepeatModeler2 and EDTA in identifying more full-length TE families.

(2) **The signature-based method:** Both RepeatModeler2, EDTA, and HiTE use LTR_retriever for detecting LTR transposons. However, only RepeatModeler2 and HiTE develop LTR redundancy removal methods. The distinction lies in their implementation: RepeatModeler2 uses MAFFT and Ninja to score gaps as a single difference, then uses Refiner to generate consensus sequences. On the other hand, HiTE begins with BLASTN for all-vs-all comparisons, applies the FMEA algorithm to bridge gaps, clusters sequences that bridge gaps, conducts multiple sequence alignment using MAFFT, and generates consensus sequences based on the majority rule.

To our knowledge, RepeatModeler2 has not designed signature-based modules specifically for TIR, Helitron, and non-LTR transposons. EDTA develops the identification modules of TIR and Helitron transposons. HiTE also develops separate modules (HiTE-TIR, HiTE-Helitron, and HiTE-NonLTR) to detect TIR, Helitron, and non-LTR transposons with fine-grained boundaries. As shown in Supplementary Tables 5-7, since the signature-based method of HiTE (HiTE-NoFiltering) uses all potential TE candidate fragments generated by our FMEA algorithm as its input, it achieves higher performance compared to EDTA (EDTA-NoFiltering).

(3) **The filtering method:** The main difference in the filtering methods used by HiTE, RepeatModeler2, and EDTA is that HiTE develops a novel homology-based filtering method based on multiple sequence alignment. This approach not only screens out many false-positive sequences but also dynamically identifies genuine TE boundaries. As shown in Supplementary Tables 5-7, the filtering method of HiTE outperforms that of EDTA.

In response to Comment 1 and Supplementary Note 6, we provided a detailed explanation of the differences

between HiTE and other pipelines.

Comment 5. *“HiTE uses multiple filtering methods to ensure that the candidates discovered have complete TE structures”*

EDTA and RepeatModeler2 use lots of filtering as well. Highlight the differences.

Authors' Response. Thank you for the comment. RepeatModeler2 makes use of the parameter '*LTRMaxSeqLen*' to set a limit on the maximum length of internal sequences. Additionally, RepeatModeler2 employs TRF to mask tandem repeats. Furthermore, LTR_retriever is employed for the identification of LTR transposons, which play a pivotal role as the primary reference during the integration of results from RECON and RepeatScout. As shown in Supplementary Figs. 4-6, we outline the main differences among the filtering methods employed by EDTA, LTR_retriever, and HiTE.

The primary advantage of the filtering method in HiTE lies in its development of a novel homology-based filtering method based on multiple sequence alignment. This method not only filters out false-positive sequences, but also dynamically identifies genuine TE boundaries. As shown in Supplementary Tables 5-7, the filtering method of HiTE shows a higher performance over that of EDTA.

In our response to Comment 1 and Supplementary Note 6, we provide a detailed explanation of the differences in the filtering method between HiTE and other pipelines.

Comment 6. *“As shown in Supplementary Table 2, we notice that HiTE exhibits relatively lower sensitivity. To explore further, we conduct a manual analysis of the missed TIR transposons in Arabidopsis thaliana.”*

Supplementary Table 2 is large with a lot of numbers and acronyms. It's difficult to parse out exactly what the authors are referring to when they state that there is “relatively lower sensitivity”. Also, what does this mean? Is this compared to other pipelines?

Authors' Response. Thanks for pointing this out. To enhance clarity and simplicity, we have divided the original Supplementary Table 2 into two separate tables (Supplementary Tables 2 and 3) in the revised Supplementary text. In Supplementary Table 2, we give the results of general-purpose TE annotators using different genomes based on the benchmarking method of RepeatModeler2 (BM_RM2). In Supplementary Table 3, we give the performance of general-purpose TE annotators using different genomes based on the benchmarking method of EDTA (BM_EDTA).

As shown in Supplementary Table 3, due to the exclusion of numerous fragmented sequences lacking clear structures or multiple-copy support, HiTE exhibits relatively lower sensitivity compared to other pipelines when using BM_EDTA. However, since BM_EDTA calculates all bases with overlaps against the gold standard annotations, it potentially leads to misleadingly high sensitivity. Therefore, it may not be the most suitable method for evaluating high-quality full-length TE libraries.

Hence, we have designed a new benchmarking method, BM_HiTE. As shown in Supplementary Fig. 2, a sequence is considered a true positive only when the length of the overlap between the test and Repbase TE sequence exceeds 95% of their lengths respectively. All other alignments, including those with significant shifts, longer sequences containing the true TE, and fragmented TE sequences, are regarded as false positives. As shown in Supplementary Table 4, HiTE exhibits superior performance in terms of both precision and sensitivity

compared to other pipelines using BM_HiTE, with thresholds of 0.95 and 0.99 for length coverage, respectively.

We have added this analysis in detail in Supplementary Notes 2, Supplementary Fig. 2, and the section '*HiTE accurately detects more intact TE models*' in the revised manuscript.

Comment 7. *Authors spend a page or two explaining how their pipeline works and the different modules. These same modules can be found in other pipelines as well. What makes this program perform better?*

Authors' Response. Thank you for your comment. HiTE distinguishes itself from existing pipelines in the design and implementation of *de novo*, signature-based, and filtering methods. The integration of all these methods allows HiTE to achieve higher performance.

The *de novo* method: In contrast to the existing pipeline, HiTE develops a novel fault-tolerant mapping expansion algorithm, denoted as FMEA, to identify the full-length TEs. Our FMEA algorithm is not primarily concerned with identifying precise TE families. Instead, it is designed to connect and span insertions and gaps between alignment segments, aiming to preserve the integrity of TE instances as much as possible.

The FMEA algorithm is the *de novo* TE searching module of HiTE, which provides the input for the subsequent signature-based TE searching module. As shown in Supplementary Table 2, the FMEA algorithm outperforms other pipelines in identifying more full-length TE families.

The signature-based method: Unlike existing signature-based methods such as EDTA, which use the whole genome as input, HiTE performs identification on all potential TE candidate fragments generated by our FMEA algorithm. As shown in Supplementary Fig. 3(B), given the misleading nature of TE structural signals, the signature-based methods using the whole genome as input often lead to more false positives.

As shown in Supplementary Table 5, under the same conditions without the use of filtering algorithms, the signature-based method of HiTE (HiTE-NoFiltering) can identify more perfect TE families compared to the signature-based method of EDTA (EDTA-NoFiltering) when using the benchmarking method of RepeatModeler2 (BM_RM2). When using the benchmarking method of EDTA (BM_EDTA), HiTE-NoFiltering shows comparable or higher performance compared to EDTA-NoFiltering (Supplementary Table 6).

However, since BM_EDTA calculates all bases with overlaps against the gold standard annotations, it may not be the most suitable method for evaluating high-quality full-length TE libraries. To better assess the quality of the test library, we have designed a new benchmarking method, BM_HiTE (Supplementary Fig. 2). As shown in Supplementary Table 7, HiTE-NoFiltering exhibits higher sensitivity and precision compared to EDTA-NoFiltering when evaluated using BM_HiTE.

The filtering method: Both EDTA and HiTE use various filtering methods (Supplementary Figs. 4 and 5). Among these methods, a flanking sequence-based filtering method employed by EDTA, LTR_retriever, and HiTE is particularly crucial (Supplementary Fig. 6). The flanking sequence-based filtering method of HiTE differs from those of EDTA and LTR_retriever in that we have developed a novel homology-based filtering method based on multiple sequence alignment. This method not only filters out many false-positive sequences but also dynamically identifies genuine TE boundaries. As shown in Supplementary Tables 5-7, the filtering method of HiTE shows a substantial superiority over that of EDTA.

We have added this analysis in detail in Supplementary Note 6, Supplementary Fig. 3, and the '*Overview of HiTE*' section in the revised manuscript.

Comment 8. *Do the authors have the necessary rights to use the Repbase library?*

Authors' Response. Thank you for the question. KAUST (King Abdullah University of Science and Technology), Saudi Arabia, holds the rights to download and utilize Repbase. According to Repbase's Academic User Agreement, Repbase can be shared and utilized within research groups.

One of our co-authors, Professor Xin Gao, a distinguished expert in Bioinformatics and AI, serves as the Director of the Computational Bioscience Research Center (CBRC), Deputy Director of Smart Health Initiative, and head of the Structural and Functional Bioinformatics Group at KAUST, has been a long-term research collaborator with our team. Together, in both 2021 and 2022, we co-authored substantial research contributions on repeat region detection published in the *Nucleic Acids Research* journal [1-2]. Throughout our collaboration, we facilitated the sharing of all research data. Please note that our work only involves downloading and testing our method on Repbase, but does not provide secondary use of Repbase data or include the data in any other database made available through our work.

References

- [1] Liao, Xingyu, Min Li, **Kang Hu**, Fang-Xiang Wu, **Xin Gao**, and **Jianxin Wang**. "A sensitive repeat identification framework based on short and long reads." *Nucleic Acids Research* 49, no. 17 (2021): e100-e100.
- [2] Liao, Xingyu, **Kang Hu**, Adil Salhi, You Zou, **Jianxin Wang**, and **Xin Gao**. "msRepDB: a comprehensive repetitive sequence database of over 80 000 species." *Nucleic acids research* 50, no. D1 (2022): D236-D245.

Comment 9. “ *Therefore, we can only evaluate HiTE and EDTA separately based on their recognition modules for different types of transposons.* ”

RepeatModeler2 contains an optional LTR detection parameter.

Authors' Response. Thank you for your comment and suggestion. RepeatModeler2 can perform LTR detection with the '*-LTRStruct*' parameter. In this study, all experimental results for RepeatModeler2 have been conducted using this parameter. Since RepeatModeler2, EDTA, and HiTE all use LTR_retriever for LTR retrotransposon identification, our goal in this context is to compare the performance of various tools in identifying two types of transposons: TIR and Helitron. To address the reviewer's concern, we have generated TIR, Helitron, and Non-LTR libraries from the output of RepeatModeler2 using the classification labels, and these libraries have been added to the evaluation results of different TE modules (Supplementary Tables 5, 6, 7, 9, 10, and 11).

As shown in Supplementary Tables 5 and 9, when using the benchmarking method of RepeatModeler2 (BM_RM2), HiTE identifies more perfect TE families compared to RepeatModeler2. When using the benchmarking method of EDTA (BM_EDTA), HiTE shows comparable performance compared to RepeatModeler2 (Supplementary Tables 6 and 10). Since BM_EDTA calculates all bases with overlaps against the gold standard annotations, it may not be the most suitable method for evaluating high-quality full-length TE libraries. To better assess the quality of the test library, we have designed a new benchmarking method, BM_HiTE (Supplementary Fig. 2). As shown in Supplementary Tables 7 and 11, HiTE exhibits higher performance compared to RepeatModeler2 when evaluated using BM_HiTE.

Comment 10. *“It is important to note that although the results of RepeatModeler2 may appear to be categorized, we cannot directly extract different types of transposons based on the classification labels, as the classification module relies on known TE databases to annotate the identification results, thereby introducing prior information.”*

Earlier in the text it was stated that the classification module of RepeatModeler2 was utilized for classification purposes in this pipeline. If it's not accurate, based on the reasons you indicate, why have you chosen to use it for your pipeline?

Authors' Response. Thank you for the question. In the revised manuscript and supplementary text, we have obtained TIR, Helitron, and Non-LTR libraries from the output of RepeatModeler2 using the classification labels, and these libraries have been added to the evaluation results of different TE modules (Supplementary Tables 5, 6, 7, 9, 10, and 11).

We consider TE identification and classification as different stages. Employing homology for TE classification based on sequence similarity is a widely adopted method. Our testing shows that RepeatClassifier, which relies on homology searches, consistently achieves high accuracy in many cases. Therefore, we use RepeatClassifier to classify identified TE families.

Comment 11. *“ Moreover, HiTE-Helitron identifies longer TE models and generates a considerably smaller library in comparison to HiTE-Helitron-NoFiltering.”*

Is this a reduction in redundancy?

Authors' Response. Thank you for the question. The HiTE-Helitron module not only addresses redundancy issues but also removes false positives, including candidate Helitron elements with inaccurate 5'-TC ends. HiTE-Helitron-NoFiltering uses EAHelitron and HelitronScanner to search for hairpin loop structures within all potential TE candidate fragments generated by our FMEA algorithm. However, due to the weak structural features of 5'-TC ends in Helitrons, both EAHelitron and HelitronScanner face challenges in accurately determining the true 5'-TC ends, resulting in numerous false positives.

HiTE-Helitron identifies the copies of candidates with flanking regions on the genome, and then applies our homology-based filtering method based on multiple sequence alignment to identify the accurate boundaries of Helitrons. This process screens out candidates with incorrect 5'-TC ends and excludes Helitron candidates that do not insert at 'AT' sites. Finally, consensus sequences are generated based on the results of multiple sequence alignment.

As shown in Supplementary Tables 5-7, HiTE shows a substantial improvement compared to HiTE-NoFiltering. This improvement is characterized by a reduction in the presence of fragmented sequences (*Good* and *Present*) and an increase in precision. We have conducted a thorough analysis, outlined in Supplementary Note 4 and the '*HiTE-TIR/Helitron detects TE models with fine-grained boundaries*' section in the revised manuscript.

Comment 12. “ We developed structural-based methods for TE identification, which exploit the characteristic features of TEs, such.....”

While this pipeline filters the data, there is no development of unique tools to handle various TE types.

Authors' Response. Thank you for your comment. HiTE can detect various types of transposons, including LTR, TIR, Helitron, and Non-LTR. In contrast to the existing pipeline, HiTE develops a novel fault-tolerant mapping expansion algorithm, denoted as FMEA, to identify the full-length TEs. The FMEA algorithm is the *de novo* TE searching module of HiTE, which provides the input for the subsequent signature-based TE searching module. As shown in Supplementary Table 2, the FMEA algorithm outperforms other pipelines in identifying more full-length TE families.

(i) For the identification of LTR transposons. Although HiTE uses the existing tool LTR_retriever for the identification of LTR-RTs, it also designs specialized methods to mitigate redundancy. This is necessary due to the frequent deletions observed within LTR transposons. HiTE begins with BLASTN for all-vs-all comparisons, applies our FMEA algorithm to bridge gaps, clusters sequences that bridge gaps, performs multiple sequence alignment using MAFFT, and ultimately leads to the generation of consensus sequences based on the majority rule.

(ii) For the identification of TIR and Helitron transposons. HiTE has developed the HiTE-TIR and HiTE-Helitron modules tailored for these two types of transposons. Unlike existing methods that use the whole genome as input, HiTE performs identification on all potential TE candidate fragments generated by our FMEA algorithm. As shown in Supplementary Tables 5 and 7, HiTE-TIR and HiTE-Helitron modules show higher performance compared to existing tools.

(iii) For the identification of non-LTR transposons. We have developed the HiTE-NonLTR module specifically designed for identifying non-LTR transposons, which searches for target site duplications (TSDs) and polyA/T tails for non-LTR transposons based on the outputs of our FMEA algorithm. As shown in Supplementary Tables 9 and 11, HiTE-NonLTR module outperforms existing tools. Due to the variability and lack of discernible structural signals of non-LTR elements, we have developed a homology-based non-LTR searching module called HiTE-NonLTR-homology to achieve high-precision non-LTR annotation (Supplementary Tables 9-11).

We have added this analysis in detail in response to Comment 1 and Supplementary Note 6.

Methods

Comment 13. *What are the sizes of the “genome chunks” during first step?*

Authors' Response. Thank you for your attention to detail. The default size of "genome chunks" in HiTE is 400 MB, and users can customize it by using the parameter "*--chunk_size*". As shown in Supplementary Fig. 11(a), slicing the genome leads to the loss of some low-copy and scattered TEs, resulting in a decrease in perfect TE families. Considering larger *chunk_size* requires more computational resources, HiTE sets *chunk_size* to 400 MB by default.

Comment 14. *I cannot properly access your code. Please send other file format.*

Authors' Response. We have recently updated the HiTE code to version 3.0. The code for HiTE is well documented on GitHub (<https://github.com/CSU-KangHu/HiTE>), which includes the source code and three different installation methods (singularity, docker, and conda). To assist the reviewers in reading and understanding our code, we have also provided an explanation of the code's structure (<https://github.com/CSU-KangHu/HiTE#code>). Additionally, we offer a Zenodo link for downloading HiTE 3.0 (<https://zenodo.org/records/10158451>).

Comment 15. *What is the output type of your program? E.g., fasta sequences, bed files, GFF, etc.*

Authors' Response. Thank you for the question. HiTE generates TE library in “fasta” sequence format. In addition to a comprehensive library encompassing all types of TEs, we also create separate libraries for different types of TEs. This allows users to utilize a particular module result of HiTE. Additionally, if the user specifies the “--annotate 1” parameter, we also use RepeatMasker for genome annotation based on the identified TE library, resulting in “.out”, “.gff”, and “.tbl” files. Please refer to <https://github.com/CSU-KangHu/HiTE#outputs> for further details.

Comment 16. *Where/what are the intermediate files? This is to ensure that the provenance of the models is maintained and fosters reproducibility as well as understanding the data.*

Authors' Response. Thank you for the question and suggestion. HiTE allows users to specify the output directory using the “--outdir” parameter. The output directory contains all intermediate files. For instance, the files “confident_TE.cons.fa” and “confident_TE.cons.fa.classified” correspond to the unclassified and classified TE libraries generated by HiTE, respectively. Additionally, the files “longest_repeats_*.fa” are outputs produced by the FMEA algorithm. Meanwhile, the files “confident_tir_*.fa”, “confident_helitron_*.fa”, and “confident_non_ltr_*.fa” represent the identification results of HiTE's TIR, Helitron, and non-LTR modules, respectively. Moreover, the files “confident_other_*.fa” denote the identification results of the homology-based non-LTR searching module. For further details, please refer to <https://github.com/CSU-KangHu/HiTE#outputs>.

In case of unexpected termination during program execution, users can resume from the last failed step using the “--recover 1” parameter. If users wish to retain all detailed output files, they need to specify the “--debug 1” parameter, which will preserve all temporary files.

Comment 17. *Which consensus caller does the pipeline use? This can greatly affect the results.*

Authors' Response. Thank you for the question. We use different consensus calling methods for various TE types:

(1) The LTR retrotransposons: Due to the presence of extensive deletions in LTR transposons, clustering methods like CD-HIT-EST struggle with redundancy removal. To address this, we conduct all-vs-all BLASTN alignments based on the results obtained from LTR_retriever. Subsequently, we apply our FMEA algorithm to bridge the gaps caused by large deletions and cluster sequences that span these gaps. We then perform multiple sequence alignment using MAFFT and generate a consensus sequence based on majority rule.

(2) The TIR and Helitron transposons: We search for the copies of TE instances, and use MAFFT based on the majority rule for consensus calling.

(3) The Non-LTR retrotransposons: Given the frequent variability in tail lengths even among elements of the

same family, we select the longest copy from the results of MAFFT as the representative sequence.

We employ CD-HIT-EST with the parameter “*-aS 0.95 -aL 0.95 -c 0.8 -G 0 -g 1 -A 80*” for clustering all types of TEs. Since most TE instances identified by HiTE are full-length, we opt for a more stringent rule of 95-95-80-80, as opposed to the conventional 80-80-80 rule. We have added this explanation in detail in the ' *Generation of a classified TE library* ' section of the revised manuscript.

Comment 18. *Your RepeatMasker settings:*

```
RepeatMasker -lib rebase_lib -nolow -pa threads test_lib
```

will miss older families, as the search stages associated with the -lib option are greatly reduced as compared to the -species option. In addition, adding the -nolow option will increase the number of false positives in your output.

Authors' Response. Thank you for your attention to detail and suggestion. In our evaluation, we use RepeatMasker to perform the benchmarking method of RepeatModeler2 (BM_RM2). The command we use is sourced from the documentation recommended by RepeatModeler2 (https://github.com/jmf422/TE_annotation/blob/master/get_family_summary_paper.sh).

This command is used to evaluate the differences between the Rebase and test library, producing metrics such as *Perfect*, *Good*, and *Present*. Since we do not conduct genome annotation using the Dfam library of RepeatMasker, we opt for the “*-lib*” option instead of “*-species*”. Similarly, in our evaluation of test TE libraries, we use the “*-nolow*” parameter to avoid masking low-complexity DNA or simple repeats.

To address the reviewer's concern, we performed the BM_RM2 experiment without the “*-nolow*” parameter as suggested (Table R1). The experimental results indicate that removing the “*-nolow*” parameter causes the test library to align with the simple repeats included in the RepeatMasker library. This leads to an increase in *Good* and *Present* metrics, while *Perfect* slightly decreases.

Table R1 The difference in performance for the benchmarking method of RepeatModeler2 (BM_RM2) after excluding the “*-nolow*” parameter. RS: RepeatScout; RM2: RepeatModeler2; Pe: Perfect; Go: Good; Pr: Present; NF: Not_found.

Species	Tools	BM_RM2 (-nolow)				BM_RM2 (without “ -nolow ”)			
		Pe	Go	Pr	NF	Pe	Go	Pr	NF
O. sativa	RS	171	139	359	2,379	180	193	483	2,192
	EDTA	514	273	863	1,398	507	573	1,713	255
	RM2	378	104	366	2,200	374	146	461	2,067
	HiTE	909	94	363	1,682	897	161	433	1,557

Comment 19. *Neither the main nor supplementary text contains any references.*

Authors' Response. Thank you for bringing this up. In the original manuscript, we incorporated a total of 68 references, ranging from page 33 to page 37. For the revised manuscript, there are 67 references, now spanning from page 36 to page 40. In the original supplementary text, there were 3 references included, situated on page 24. For the revised supplementary text, there are 6 references, which can be found on page 35.

Answers to Reviewer #2

Reviewer #2 (Remarks to the Author):

Comment 1. *Hu et al. present the development of HiTE for fast and accurate annotation of full-length TEs in both plant and animal genomes. I find this topic of high importance since sequencing and assembly of eukaryotic genomes have been much improved, yet genome annotation is still lagging with slow speed and inaccurate results. The proposed approach is very promising, showing much-improved precision and full-length TE detection. However, the sensitivity seems to be over-sacrificed to achieve high precision, resulting in an overall lower sensitivity than companion tools. TIR detection of HiTE is very promising, yet with the lack of further classification into clades. The execution speed of the program still has room to improve to claim as “fast.” Overall, I think HiTE is very promising but may need further improvement to become a useful tool for the genome annotation community.*

Authors' Response. We thank the reviewer for the positive and supportive comments. We have provided point-by-point responses to the reviewer's comments:

The issue of lower sensitivity in HiTE: Using the homology annotation tool RepeatMasker, full-length TE families can annotate both complete and fragmented copies that share homology with these full-length TE families in the genome. Therefore, an ideal TE library should exclusively consist of intact TE families, without any fragments [1, 2]. However, the genome contains many fragmented copies of full-length TEs, and their proportion even surpasses that of full-length TEs. The fragmented TE sequences in the test library can be aligned to these fragmented copies of full-length TEs, which can cause misleadingly high sensitivity. At the same time, the benchmarking method of EDTA (BM_EDTA) calculates all bases with overlaps against the gold standard annotations, it also leads to misleadingly high sensitivity. Therefore, BM_EDTA may not be the most suitable method for evaluating high-quality full-length TE libraries (Supplementary Note 2). As shown in Supplementary Table 3, due to the exclusion of numerous fragmented sequences lacking clear structures or multiple-copy support, HiTE exhibits relatively lower sensitivity compared to other pipelines when using BM_EDTA.

To address this issue, we have designed a new benchmarking method, referred to as BM_HiTE (Supplementary Fig. 2). It filters out the fragmented annotation and retain only the full-length TE copies. The major difference between BM_HiTE and BM_EDTA lies in the criteria for calculating true positives. BM_EDTA considers any sequence that has any length of match with the standard library as a true positive. In BM_HiTE, a sequence is considered a true positive only when the length of the overlap between the test and Repbase TE sequence exceeds 95% of their lengths respectively. As shown in Supplementary Fig. 2, all others are considered as false positives, including alignments with significant shifts, longer sequences containing the true TE, and fragmented TE sequences. The inspiration for the 95% length coverage threshold comes from BM_RM2. However, such a threshold is still considered inaccurate and misleading [3]. Therefore, we have employed two distinct thresholds, namely the rough perfect definition (greater than 95% length coverage) and the precise perfect definition (greater than 99% length coverage), to assess the performance of different tools. As shown in Supplementary Table 4, compared to existing methods, HiTE exhibits higher performance in terms of both precision and sensitivity when using BM_HiTE.

The issue of unclassified elements in HiTE: The identification and classification of TEs are two different stages. HiTE focuses on TE identification within the genome. For TE classification, we use RepeatClassifier (a component of RepeatModeler2) to classify the identified transposons. The performance of RepeatClassifier depends on the homology search library of RepeatMasker. The default installation of RepeatMasker through conda only includes a curated library, requiring additional configuration for a complete library to achieve

optimal classification performance. Please note that this configuration step is specifically necessary for conda installation (refer to https://github.com/CSU-KangHu/HiTE#install_conda). For docker and singularity installations, we have already integrated the full libraries, eliminating the need for additional steps.

The issue of HiTE running slowly on large genomes: In the latest version of HiTE 3.0, we implement a "*mask-identify-mask...*" computation pattern, referred to as the pre-masking step. Users can specify whether to employ this step using the parameter "*--is_prev_mask*". Given that TEs are repetitive elements with numerous copies in the genome, this pre-masking step significantly reduces redundant computations. As shown in Supplementary Table 13, compared to the original version of HiTE, employing the pre-masking step reduces runtime for large genomes like *D. rerio* and *Z. mays*, requiring only 3.5 and 23.5 hours, respectively. Additionally, we have conducted tests on the human genome using 20 threads, HiTE using the pre-masking step requires only 21.5 hours.

Reference:

- [1] Storer JM, Hubley R, Rosen J, Smit AF. Methodologies for the de novo discovery of transposable element families. *Genes*. 2022 Apr 17;13(4):709.
- [2] Flynn JM, Hubley R, Goubert C, Rosen J, Clark AG, Feschotte C, Smit AF. RepeatModeler2 for automated genomic discovery of transposable element families. *Proceedings of the National Academy of Sciences*. 2020 Apr 28;117(17):9451-7.
- [3] Rodriguez, M. & Makałowski, W. Software evaluation for de novo detection of transposons. *Mobile DNA* 13, 1-14 (2022).

Major

Comment 2. *RepeatModeler2 has pretty consistent performance in terms of execution time, which could benefit large genome annotation. For example, RM2 spent 24 hours in maize, while HiTE took over 32 hours. The reviewer tested on slightly larger genomes, such as human and maize, the HiTE ran very slowly compared to other programs. The author claims this is a fast program in their title but testing on the human genome costed almost 10 days using 20 CPU threads, which is not as fast as I thought it should be.*

Authors' Response. Thank you for bringing this up. In the latest version of HiTE 3.0, there are two main modifications which have a close relation to the execution time. (1) We have added EAHelitron to identify more Helitron elements using the outputs of our FMEA algorithm, thus requiring more runtime compared to the original version of HiTE. For instance, the original version of HiTE took 32.5 hours in maize, whereas HiTE now takes 70.5 hours. (2) To make HiTE faster, we implement a "*mask-identify-mask...*" computation pattern, referred to as the pre-masking step. Users can specify whether to employ this step using the parameter "*--is_prev_mask*". To minimize computational costs during a single run, we partition the large genome assembly into smaller chunks. Before iteratively detecting TEs for each new genome chunk, we first mask the chunk using the identified TEs. Given that TEs are repetitive elements with numerous copies in the genome, this pre-masking step significantly reduces redundant computations.

HiTE using the pre-masking step requires less running time compared to the original version of HiTE. We have conducted runtime tests for EDTA, RepeatModeler2, and HiTE on the human genome using 40 threads, which have taken 28.5, 16.5, and 15.8 hours respectively. We have also tested HiTE on the human genome with 20 threads, which has taken 21.5 hours. On maize, EDTA has taken 62 hours using 40 threads, while RepeatModeler2 and HiTE have only taken 24 hours.

We have conducted a performance comparison of HiTE before and after the application of the pre-masking step. As shown in Supplementary Table 12, the application of the pre-masking step has tended to result in a decrease in metrics, including perfect TE families, sensitivity, and precision. This decrease may be attributed to the inclusion of fragmented or false-positive sequences in the previous round, which could lead to the partial masking of genuine TEs in the next round, rendering them unidentifiable.

As shown in Supplementary Tables 2 and 4, despite the potential decrease in metrics when using the pre-masking step, HiTE still outperforms existing methods. Therefore, HiTE defaults to enabling the pre-masking step and the results shown in our paper are tested by using the default except those marked with “un-pre-mask”. However, if users prefer a more comprehensive TE library and are willing to accept longer runtimes, they can disable the pre-masking step by setting “`--is_prev_mask 0`”.

Comment 3. *Suppl. Table 2. Why is HiTE-FMEA only available in rice? It seems to me it's still an experimental module that is not readily available on other species.*

Authors' Response. Thank you for your attention to detail. HiTE-FMEA, a module of HiTE responsible for *de novo* TE searching, identifies coarse-grained TE candidates, forming the foundation for subsequent fine-grained TE detection (Supplementary Note 6). This module is crucial and applicable across all species.

However, since HiTE-FMEA primarily focuses on generating coarse-grained TE candidates, it encompasses numerous false positives, fragments, and redundant sequences. As shown in Supplementary Fig. 8, HiTE-FMEA has generated a substantial output size of 112 Mb candidates in rice. Such extensive results can lead to evaluation programs failing due to memory constraints or requiring a long evaluation time.

To address the reviewer's concern, we have expanded HiTE-FMEA evaluations to include other species. In Supplementary Table 2, HiTE-FMEA has identified many full-length (*Perfect*) and fragmented (*Good* and *Present*) TE families. As shown in Supplementary Table 3, when using BM_EDTA, HiTE-FMEA has achieved higher sensitivity but lower precision compared to other tools. However, it is important to note that BM_EDTA calculates all bases with overlaps against gold standard annotations, which may result in misleadingly high sensitivity. As shown in Supplementary Table 4, HiTE-FMEA shows lower precision compared to other tools when using BM_HiTE. This underscores the significance of developing signature-based and filtering methods to mitigate false positives and fragmented sequences within the outputs of our HiTE-FMEA module. As shown in Supplementary Tables 3 and 4, due to the extensive output size of HiTE-FMEA, RepeatMasker cannot align them to large genomes like *D. rerio* and *Z. mays* because of the limitation of memory, leading to a lack of results in these cases.

Comment 4. *Although it is used in the RepeatModeler benchmarking, it is still inaccurate and misleading to call matches with >95% similarity and >95% coverage “Perfect,” which is defined as “having all the required or desirable elements, qualities, or characteristics; as good as it is possible to be.” Many structural tools search for perfect intact TEs to make high-quality libraries, such as LTR_retriever, but the definition used in this study is rough, and applying such metric on tools that are more stringent on the criteria or use a totally different approach to identify repeats would result in misleading results. For example, RepeatModeler2 uses an all-by-all blast approach to identify repeat units and is not designed to identify “perfect” intact TEs (Fig. 2a).*

Authors' Response. Thank you for the suggestion, which we fully agree. In the benchmarking method of RepeatModeler2, a sequence is considered “*Perfect*” if it meets the criteria of >95% similarity, >95% coverage, and <5% divergence with the gold standard sequences. While this definition provides a general assessment of

the presence of full-length TE families in the test library, it is admittedly a coarse measure. Given that some level of divergence is acceptable within a TE family, we maintain the criteria of >95% similarity and <5% divergence while making the length coverage more stringent, changing it to >99%. We refer to ">95% and >99% length coverage" as the rough and precise perfect definitions, respectively.

As shown in Supplementary Table 2, HiTE identifies more perfect TE families under both the rough and precise perfect definitions compared to existing tools. We have conducted similar tests using our proposed benchmarking method of HiTE (BM_HiTE) with the stricter >99% threshold of length coverage (Supplementary Table 4). The experimental results indicate that even with a more stringent threshold, HiTE still outperforms other tools.

RepeatModeler uses an all-by-all blast approach to identify repeat units. However, the original version of RepeatModeler, like other existing TE discovery software, falls short of producing a complete, nonredundant library of full-length consensus sequences [1]. In the improved version of RepeatModeler2, it uses LTR_retriever to identify LTR-RTs, which serve as the primary reference in integrating results from RECON and RepeatScout. RepeatModeler2 enhances the number of perfect TE families and improves the quality of the TE library compared to the original version of RepeatModeler.

References

[1] Flynn JM, Hubley R, Goubert C, Rosen J, Clark AG, Feschotte C, Smit AF. RepeatModeler2 for automated genomic discovery of transposable element families. *Proceedings of the National Academy of Sciences*. 2020 Apr 28;117(17):9451-7.

Comment 5. *“However, the evaluation of BM_EDTA relies on the count of matched bases instead of the complete TE sequence, potentially leading to the inclusion of short false-positive sequences or fragments, which in turn can cause an erroneous elevation of sensitivity.” As far as I know, EDTA has two approaches to annotating TEs. The first approach uses structural-based methods to identify full-length TEs (the “Perfect” category). The second approach uses RepeatMasker for homology annotation of fragmented TEs (The Good and Present categories). The authors refer to the homology annotation approach, which effectively suggests that RepeatMasker could include short false-positive sequences. Such a claim is not supported by their data nor by other publications, which can be viewed as an excuse for their low sensitivity results. In fact, the authors seem to be biased toward the complete TE sequence, which consists of only a small fraction of most eukaryotic genomes. Most genomes are full of fragmented TE sequences because of genome purging.*

Authors' Response. Thank you for your comment. In the first stage, EDTA, RepeatModeler2, and HiTE use different methodologies to identify full-length TEs. In the second stage, RepeatMasker serves as the gold standard tool for genome annotation, which is utilized by all three tools to annotate the genome. Therefore, what truly determines the performance of different TE identification tools lies in the diversity of full-length TE libraries generated in the first stage. RepeatMasker is a homology-based annotation tool, which means that when provided with a TE library using the "-lib" parameter, it utilizes alignment tools like RMBlast to search and annotate sequences across the entire genome that are homologous to the library. This includes both full-length and fragmented TE sequences. Therefore, an ideal library would contain only full-length models of all significantly distinct TEs that have left copies in a genome [1, 2]. The goal of HiTE is to produce a non-redundant library solely comprising full-length TEs.

The inclusion of fragmented sequences in the TE library would hamper the classification of the TE families, inflate the number of actual TE families in the genome, and confound genome annotation and downstream

analyses [2]. However, the genome contains many fragmented copies of full-length TEs, and their proportion even surpasses that of intact full-length TEs. The fragmented TE sequences in the test library can be aligned to these fragmented copies of full-length TEs, causing misleadingly high sensitivity. At the same time, the benchmarking method of EDTA (BM_EDTA) calculates all bases with overlaps against the gold standard annotations, it may not be the most suitable method for evaluating high-quality full-length TE libraries.

To better assess the quality of the test library, we have designed a new evaluation approach, referred to as BM_HiTE, as shown in Supplementary Fig. 2. The main difference between BM_HiTE and BM_EDTA lies in the stricter definition of "true positive". A sequence is considered a true positive only when the length of the overlap between the test and Repbase TE sequence exceeds 95% of their lengths respectively. All others are considered as false positives, including alignments with significant shifts, longer sequences containing the true TE, and fragmented TE sequences.

As shown in Supplementary Table 4, HiTE exhibits higher performance in terms of both precision and sensitivity compared to other pipelines when using BM_HiTE.

Reference:

- [1] Storer JM, Hubley R, Rosen J, Smit AF. Methodologies for the de novo discovery of transposable element families. *Genes*. 2022 Apr 17;13(4):709.
- [2] Flynn JM, Hubley R, Goubert C, Rosen J, Clark AG, Feschotte C, Smit AF. RepeatModeler2 for automated genomic discovery of transposable element families. *Proceedings of the National Academy of Sciences*. 2020 Apr 28;117(17):9451-7.

Comment 6. *In rice, the non-filtered Helitorns identified by HiTE have pretty high sensitivity (90%), but when filtering is applied by HiTE, it has low sensitivity (61.3%), suggesting the filtering criteria are not very specific. In Arabidopsis, the trend is similar, except the non-filtered Helitrons by HiTE have quite a low sensitivity (70.7%), resulting in a further lower sensitivity when filtered by HiTE (55.4%). See Fig 4e. Again, the sensitivity performance is suboptimal in HiTE, which is critical for genome annotations.*

Authors' Response. Thank you for bringing this up. As shown in Supplementary Table 6, due to the exclusion of numerous fragmented sequences, the sensitivity of HiTE has decreased after applying our filtering method when using BM_EDTA. Compared to BM_EDTA, our newly proposed evaluation method, BM_HiTE, mitigates the impact of fragmented sequences on the results, making it more suitable for evaluating high-quality full-length TE libraries (Supplementary Note 2).

As shown in Supplementary Table 7, under normal circumstances, applying filtering methods tends to result in a decrease in sensitivity. For instance, when EDTA applies filtering, although precision improves, sensitivity simultaneously decreases. However, we observe that the sensitivity of HiTE after filtering often exceeds the sensitivity before filtering in many cases (Supplementary Tables 7 and 8). This is attributed to the fact that our filtering algorithm not only screens out false positives but also dynamically identifies TE boundaries, resulting in the transformation of TE families originally categorized as "Good" or "Present" into "Perfect".

We have observed this phenomenon across multiple species, including *O. sativa*, *A. thaliana*, and *C. briggsae*. This highlights the effectiveness of our filtering method. We have added the analysis in Supplementary Note 4 and the 'HiTE-TIR/Helitron detects TE models with fine-grained boundaries' section of the revised manuscript.

Comment 7. *Non-LTR identification relies on existing libraries of the species to be annotated, which is a limiting factor in the application of HiTE on non-model species without existing non-LTR libraries. In species without manually curated SINE/LINE libraries, the annotation sensitivity of HiTE will be very low, if not 0.*

Authors' Response. Thank you for your comment and suggestion. Reliable *de novo* identification of non-LTR elements is difficult due to their inherently challenging recognition characteristics. The most effective method for non-LTR identification involves searching known non-LTR libraries and relying on extensive expert knowledge for manual curation.

EDTA conducted tests on various non-LTR identification methods in their article. These methods include database-based homology annotation approaches like Repbase and SINEBase, as well as *de novo* detection methods such as RepeatModeler, MGEScan3, SINE-Finder, and SINE_Scan. The results in their paper show that RepeatModeler has obtained the best performance among all *de novo* methods for non-LTR identification.

To equip HiTE with the ability to *de novo* identify non-LTR elements, we have searched for target site duplications (TSDs) and polyA/T structures within the coarse-grained TE candidates generated by our HiTE-FMEA module. Consequently, we have developed a *de novo* identification module of HiTE (HiTE-NonLTR) specifically tailored for non-LTR elements. In addition, due to the variability and lack of discernible structural signals of non-LTR elements, we have also developed a homology-based non-LTR searching module called HiTE-NonLTR-homology to achieve high-precision non-LTR annotation.

As shown in Supplementary Table 10, we evaluated the entire non-LTR library, HiTE-NonLTR-homology, RepeatModeler2, and HiTE-NonLTR using BM_EDTA on the rice genome. RepeatModeler2 exhibits relatively high performance in non-LTR identification. However, RepeatModeler2 identifies only 1 perfect TE family, whereas HiTE-NonLTR identifies 18 ones (Supplementary Table 9). As shown in Supplementary Table 11, HiTE-NonLTR outperforms RepeatModeler2 using BM_HiTE. Nevertheless, when compared to a curated non-LTR library, HiTE-NonLTR still shows lower performance. Consequently, we have introduced a "--is_denovo_nonltr" parameter to control whether to use the *de novo* detection mode for non-LTR elements.

Comment 8. *Classification failed on all genomes tested on the reviewer's end. Furthermore, the reviewer used the rice result to benchmark with the EDTA lib-test.pl script, and obtained only 57.7% sensitivity on the total category (all repeats due to the lack of classification).*

Authors' Response. Thank you for bringing this up.

(1) HiTE focuses on TE identification within the genome. For TE classification, we use RepeatClassifier (a component of RepeatModeler2) to classify the identified transposons. The performance of RepeatClassifier depends on the homology search library of RepeatMasker. The default installation of RepeatMasker through conda only includes a curated library, requiring additional configuration for a complete library to achieve optimal classification performance. Please note that this configuration step is specifically necessary for conda installation (refer to https://github.com/CSU-KangHu/HiTE#install_conda).

(2) There may be two reasons why the experimental results have not been reproduced by the reviewer:

- a) The necessary tools in the "\${HiTE_Home}/tools" directory did not execute successfully. To resolve this, we have provided the command "cd \${HiTE_Home}/tools && chmod +x *" to grant the execute. We have also updated the latest README documentation on GitHub accordingly.
- b) The choice of different gold-standard libraries can influence the outcomes of the evaluation experiments.

Our commands for running these experiments are as follows:

- i. `RepeatMasker -e ncbi -pa ${threads} -q -no_is -norna -nolow -div 40 -lib ${rebase_lib} -cutoff 225 ${assembly} && mv ${assembly}.out rebase_lib.out`

- ii. `RepeatMasker -e ncbi -pa ${threads} -q -no_is -norna -nolow -div 40 -lib ${test_lib} -cutoff 225 ${assembly} && mv ${assembly}.out test_lib.out`
- iii. `perl ${EDTA_Home}/lib-test.pl -genome ${assembly} -std rebase_lib.out -tst test_lib.out -cat Total`

To facilitate the swift reproduction of experiments, HiTE includes an integrated experiment evaluation method. Once the gold-standard library is copied to the "`HiTE_Home/library`" directory, the experiments can be replicated by simply adding parameters such as "`--BM_RM2 1`" and "`--species rice`". For more detailed instructions, please refer to the tutorial provided at <https://github.com/CSU-KangHu/HiTE/wiki/Experiment-reproduction>.

Comment 9. *Precision is calculated as the proportion of true positives on all positive results, which completely ignores false negatives (suppl. Fig 1a). You can have a very high precision, meaning most of the reported TEs are true TEs, and also a very high false negative, meaning low sensitivity. The authors emphasize on high precision performance of HiTE but not so much for sensitivity because, except for TIR, other TEs have low sensitivity despite high precision.*

Authors' Response. Thank you for your comment and suggestion. Since BM_EDTA calculates all bases with overlaps against the gold standard annotations, it may not be the most suitable method for evaluating high-quality full-length TE libraries. As shown in Supplementary Fig. 2, we have designed a new benchmarking method, BM_HiTE. BM_HiTE mitigates the influence of fragmented sequences on the results, making it more suitable for evaluating high-quality full-length TE libraries. For a more detailed explanation, please refer to the response in Comment 1 and Supplementary Note 2.

As shown in Supplementary Tables 2 and 4, we have documented the performance of different tools under two length coverage thresholds: >0.95 and >0.99. HiTE detects more perfect TE families, and shows higher performance in both precision and sensitivity compared to other tools when using BM_HiTE. As shown in Supplementary Tables 7 and 11, HiTE also outperforms existing methods in the identification of various types of TEs.

In summary, after excluding the impact of fragmented sequences on the results, HiTE shows higher performance in both precision and sensitivity compared to existing tools.

Comment 10. *TIR is a superfamily of DNA transposons that contains many clades such as PILE, CMC-EnSpm, hAT, MuDR, piggyBac, TcMar_Stowaway, etc. The authors did not demonstrate HiTE's capability of classifying these clades, which is a critical defect as a comprehensive TE annotation tool.*

Authors' Response. Thank you for your comment. TE identification and classification are two different functionalities in HiTE. Currently, HiTE focuses on TE identification within the genome. For TE classification, HiTE uses RepeatClassifier (a component of RepeatModeler2) to classify identified TEs.

Minor

Comment 11. *Supp Table 1, Total is located in the middle of other TE superfamilies, which should be the last entry of each species.*

Authors' Response. Thank you for pointing this out. We have corrected this accordingly.

Comment 12. *Suppl. Tables and benchmark results for different species/categories should be separated by a line for better readability.*

Authors' Response. Thank you for the suggestion. We have added separators to all relevant supplementary tables for improved readability as suggested.

Comment 13. *Species names should be italicized consistently.*

Authors' Response. Thank you for the suggestion. We have corrected it.

REVIEWER COMMENTS

Reviewer #1 (Remarks to the Author):

I would like to take the time to thank the authors for their careful consideration of all submitted comments.

Although I was unable to access the code, the other reviewer was able to successfully utilize the code, and I have based some of my comments on their assessment, in addition to the authors' responses to my own inquiries.

Along that vein, I must thoroughly apologize for my reference and code comments. My computer opened the document using a somewhat incompatible program, not allowing for the visualization of the references. This same issue occurred when opening the code. In the future, I will make sure to reach out to the editors or authors directly in order to make sure everything is working properly.

First and foremost, I applaud the authors for being one of the first manuscripts I have read in providing not only clear rationale, but also their own benchmark and clear definitions of what a true positive is in terms of coverage and similarity. This is exactly the kind of transparency currently lacking in many TE publications, although I do recognize that the field generally struggles with generating agreed-upon standard practices. In addition, I thoroughly appreciate the additional clarity the authors provided regarding the novelty of their pipeline, as it did not come through in the first round. I am also aware that I stated that concern multiple times, which I did not fully realize at the time. Thank you for carefully considering my redundancy, a plague I am generally better at avoiding when it comes to TE discussions.

Areas of improvement:

1. It is unfortunate that the classification relies so heavily on homology to previously-discovered transposable elements. This means that the pipeline would work well for organisms that are close-related to model organisms and highly-studied genomes, but would not work well for newly-sequenced, phylogenetically-diverse species, far removed from species with known genomic characteristics. In addition, the identification of non-LTR elements would also be similarly hampered, as it is also (based on my understanding) based on homology searches as well. A recent back and forth set of publications regarding EDTA highlighted the limitations of the repeat discovery pipeline in certain species, particularly those with high amounts of non-LTR elements:

Gozashti L and Hosekstra H.E. (2024), "Accounting for diverse transposable element landscapes is key to developing and evaluating accurate de novo annotation strategies" *Genome Biology* 25:4 <https://doi.org/10.1186/s13059-023-03118-1>

Ou S., Jiang N, Hirsch C.N., and Hufford M.B. (2024), "Response to Commentary: Accounting for diverse transposable element landscapes is key to developing and evaluating accurate de novo annotation strategies" *Genome Biology* 25:6 <https://doi.org/10.1186/s13059-023-03119-0>

I am curious to see how your pipeline performs against a diverse set of genomes.

2. For the pre-masking step, it is described as finding TEs in one chunk of the genome, and then using the identified TEs to mask the next selected chunk, and so on and so forth until the genome is fully analyzed. The use of this step results in decreased perfect TE families found, sensitivity and precision. So the increase in speed comes at a loss of high quality results. Given that HiTE cannot fully process larger genome without the pre-masking step, I would recommend that this is highlighted in the discussion. While the addition of the pre-masking step does increase the speed of the program, the overall performance of the program decreases. It therefore follows that the program cannot simultaneously be both fast and accurate.

3. Please explain the findings in Figure 4c regarding the "Not Found (NF)" category. While the full-length TEs found is generally higher for HiTE than HiTE-FMEA, the NF is also much higher in HiTE. Does this mean that HiTE is missing TE families entirely as compared to Hi-FMEA? Are these novel TEs not captured in the gold standard libraries?

4. Although counterintuitive, using the "-nolow" parameter as part of RepeatMasker actually causes the TEs in the library to map to simple repeats and low complexity regions. As part of running RepeatMasker, the first and last stages are to search for young and old simple repeats, respectively. As some consensus sequences within a library, for example, Alu, contain within them simple repeat sequences, such as polyA tails and low complexity regions (GC-rich regions of Alu and LINE as examples). If these low complexity and simple repeat regions are not masked, this leads to an increase in false positives. In addition, the usage of the "-species" parameter vs. "-lib" is not specific to Dfam libraries. However, given that the same parameter was used for all species, my point is moot. While the same setting was used for all benchmarking purposes, the "-q" setting in RepeatMasker leads to less sensitive searches as compared to the default setting, or the "-s" setting, and may influence your results.

Reviewer #3 (Remarks to the Author):

I find that the explanations or analyses provided by most authors are reasonable. However, I have reservations regarding the requirement for TE annotation evaluation to be >95% identity/coverage threshold for. This evaluation actually comes from two aspects: firstly, the examination of the library itself. It is reasonable to impose stricter parameters when comparing curated libraries with de novo annotated libraries. However, since the library itself is derived from a consensus step, traditional rules such as 80-80-80 (80% identity, 80% coverage and > 80 bp) apply already introduce the difference even in the same sequence. Therefore, even if they are both full-length, there may still be some differences between them. Additionally, different clustering and filtering for nested TEs also break down the full-length TE candidates. A more significant issue arises from the uncertainty of annotation. When annotating the full-length library with Repeatmasker, all fragmented TEs can be annotated. Thus, requiring 95% coverage/identity is unreasonable. For instance, soloLTRs, which only consist of a portion of the full-length, would be excessively constrained. The authors could filter out the highly fragmented results from Repeatmasker's output, but the requirement for coverage and identity conflicts with the essence of homology-based annotation. Another concern arises from the premature masking of tandem repeats, which may result in the loss of many LTRs, such as the Dasheng LTR in rice (Jiang et.al 2002), which contains numerous internal tandem repeats. With the proliferation of T2T genomes, an increasing number of complex regions are being assembled, such as the centromere-specific ATHILA family in Arabidopsis thaliana (Naish et.al 2021) and CRM in maize. Pre-masking could also lead to missed detections of such elements although increase the speed. Can we resecure these LTR in these masking regions? Or identify them before masking the whole genome. Given that the genomes evaluated by the authors were previously assembled incompletely, it is uncertain how this step would impact organisms with T2T genomes, such as rice, Arabidopsis, or humans. Could the authors consider evaluating the extent of this impact in the discussion or as a supplement?

Thanks for your improvements on the documentation and distribution software. But I find the result still a bit messy for users. For instance, HiTE try to find full-length TE than other tools but only provides fasta. For TE biology, detailed structure of TE could be very helpful for people manually curation and comparing. Do the authors think provide a full-length TE annotation bed or gff? Like TSD, 3'LTR,5'LTR, terminal inverted repeats, hairpin loops, and polyA/T tails.

Supp Fig9, why EDTA result only contained TIR and LTR? Did the author exclude the Helitron and RepeatModeler running?

Summary

We appreciate the valuable comments and suggestions from the editor and reviewers. Based on the suggestions and comments from the editor and reviewers, we revised our paper. We addressed those comments and suggestions carefully and included a point-by-point response below. The significant changes in the revised manuscript were highlighted by red color.

Answers to Reviewer #1

Reviewer #1 (Remarks to the Author):

Comment 1. *I would like to take the time to thank the authors for their careful consideration of all submitted comments.*

Although I was unable to access the code, the other reviewer was able to successfully utilize the code, and I have based some of my comments on their assessment, in addition to the authors' responses to my own inquiries. Along that vein, I must thoroughly apologize for my reference and code comments. My computer opened the document using a somewhat incompatible program, not allowing for the visualization of the references. This same issue occurred when opening the code. In the future, I will make sure to reach out to the editors or authors directly in order to make sure everything is working properly.

First and foremost, I applaud the authors for being one of the first manuscripts I have read in providing not only clear rationale, but also their own benchmark and clear definitions of what a true positive is in terms of coverage and similarity. This is exactly the kind of transparency currently lacking in many TE publications, although I do recognize that the field generally struggles with generating agreed-upon standard practices. In addition, I thoroughly appreciate the additional clarity the authors provided regarding the novelty of their pipeline, as it did not come through in the first round. I am also aware that I stated that concern multiple times, which I did not fully realize at the time. Thank you for carefully considering my redundancy, a plague I am generally better at avoiding when it comes to TE discussions.

Authors' Response. We thank the reviewer for the positive feedback on our first-round response. We fully agree with the reviewer that the absence of a standardized benchmarking method poses a challenge in fairly evaluating different TE identification tools. Nevertheless, many researchers continue to make ongoing efforts, such as TE Hub (<http://tehub.org/>), which inspires us to continuously develop and improve HiTE with our initial intentions and motivation. Therefore, we highly value each question raised as an opportunity for continuous improvement of HiTE. We are pleased that we have addressed many concerns raised by the reviewers in the previous round. However, we must acknowledge that there is still room for improvement in HiTE. We will continue to strive for enhancement and contribute to the development of the TE community.

Areas of improvement:

Comment 2. *It is unfortunate that the classification relies so heavily on homology to previously-discovered transposable elements. This means that the pipeline would work well for organisms that are close-related to model organisms and highly-studied genomes, but would not work well for newly-sequenced, phylogenetically-diverse species, far removed from species with known genomic characteristics.*

Authors' Response. Thank you for the comment and suggestion. The identification and classification of TEs are two different stages, and HiTE focuses on TE identification within the genome. For TE classification, we employ RepeatClassifier, a homology-based search method implemented within RepeatModeler2, to classify

the detected transposons. However, using homology-based search methods yields high performance in classifying TEs from known species but exhibits low performance for classifying TEs from new species. Recently, we have developed NeuralTE (<https://github.com/CSU-KangHu/NeuralTE>), a deep learning method designed to classify transposons at the superfamily level. HiTE now supports two methods for TE classification: RepeatClassifier and NeuralTE. Users can choose between them by setting the parameter "--use_NeuralTE 1/0" to use NeuralTE or RepeatClassifier.

To achieve accurate TE classification, NeuralTE identifies various structural features of transposons, and uses different combinations of k -mers for terminal repeats and internal sequences to uncover distinct patterns. Compared to existing methods, NeuralTE is the first to integrate TSDs, 5-bp ends, domains, terminal, and internal information into a deep learning method to enhance classification accuracy. Furthermore, NeuralTE simplifies the classification process by training a single model capable of classifying different types of superfamilies. This eliminates the requirement to train multiple models for different levels of the TE hierarchical classification system, reducing both the complexity of model training and the storage demands.

We have performed benchmarking on different TE classification methods. As shown in Table R1, the results show that NeuralTE has superior performance over TERL, DeepTE, and TESorter, and comparable performance with RepeatClassifier in predicting TE classification for known transposons. When it comes to classifying novel TEs in flowering plants and mammals, NeuralTE demonstrates higher performance compared to RepeatClassifier (Tables R2 and R3).

For a more detailed overview of NeuralTE, please refer to “*Hu, K., Xu, M., Gao, X., & Wang, J. (2024). NeuralTE: an accurate approach for Transposable Element superfamily classification with multi-feature fusion. bioRxiv, 2024-01*”, which has been submitted to a conference for reviewing.

Therefore, we provide two methods, RepeatClassifier and NeuralTE, in HiTE pipeline for TE classification. Users are recommended to use RepeatClassifier when dealing with highly studied genomes and to use NeuralTE in other cases.

We have updated the TE classification module in the '*Overview of HiTE*' and '*Generation of a classified TE library*' sections of the revised manuscript.

Table R1 Evaluation of performance between different TE classification tools on known transposons.

Tools	Precision	Recall	F1-score
TERL	0.5201	0.4804	0.4893
DeepTE	0.7792	0.6338	0.6624
TESorter	0.5591	0.3273	0.3531
RepeatClassifier	0.9133	0.9022	0.9031
NeuralTE	0.9294	0.8754	0.8962

Table R2 Evaluation of different TE classification tools for predicting novel transposons in flowering plants.

Tools	Precision	Recall	F1-score
TERL	0.3659	0.3353	0.3116
DeepTE	0.3792	0.3366	0.2966
TESorter	0.6051	0.3319	0.3532
RepeatClassifier	0.7069	0.4792	0.4843
NeuralTE	0.6907	0.6855	0.6780

Table R3 Evaluation of different TE classification tools for predicting novel transposons in mammals.

Tools	Precision	Recall	F1-score
TERL	0.2188	0.2639	0.1713
DeepTE	0.2550	0.3672	0.2226
TEsorter	0.2940	0.2031	0.1209
RepeatClassifier	0.4985	0.5570	0.4270
NeuralTE	0.5421	0.6807	0.5194

Comment 3. *In addition, the identification of non-LTR elements would also be similarly hampered, as it is also (based on my understanding) based on homology searches as well. A recent back and forth set of publications regarding EDTA highlighted the limitations of the repeat discovery pipeline in certain species, particularly those with high amounts of non-LTR elements:*

*Gozashti L and Hosekstra H.E. (2024), "Accounting for diverse transposable element landscapes is key to developing and evaluating accurate de novo annotation strategies" Genome Biology 25:4
<https://doi.org/10.1186/s13059-023-03118-1>*

*Ou S., Jiang N, Hirsch C.N., and Hufford M.B. (2024), "Response to Commentary: Accounting for diverse transposable element landscapes is key to developing and evaluating accurate de novo annotation strategies" Genome Biology 25:6
<https://doi.org/10.1186/s13059-023-03119-0>*

I am curious to see how your pipeline performs against a diverse set of genomes.

Authors' Response. Thank you for your comment. We have thoroughly examined the discussions in both the commentary [1] and response [2] articles concerning EDTA. In the commentary article [1], the assessment of EDTA was conducted across various species including *Mouse*, *Zebrafish*, *Zebra finch*, and *Chicken*. In contrast to plants, animal genomes exhibit relatively lower diversity in TEs, with non-LTRs comprising the majority [3].

The identification of non-LTR elements poses a significant challenge due to their lack of prominent structural features and full-length copies. EDTA employs two approaches to address this issue. Firstly, the "--*curatedlib*" parameter is used to incorporate reliable non-LTR elements provided by users. For example, in response article [2], the inclusion of known non-LTR elements significantly enhances the annotation results of EDTA. However, EDTA requires users to provide a fully reliable non-LTR library, which is often challenging for users to achieve. In HiTE, we have developed HiTE-NonLTR-homology, which integrates a known non-LTR library within HiTE for conducting homology searches within the genome, eliminating the need for additional user intervention. Unlike EDTA, we align the known library to the genome and retain TE consensus with full-length copies within the genome, thereby obtaining a genome-specific non-LTR library.

Secondly, EDTA uses RepeatModeler2 to identify *de novo* non-LTR elements. In the original implementation of EDTA, it uses RepeatModeler2 to identify the remaining TEs that are missed by EDTA. In its recent iteration, EDTA uses TEsorter [4] to directly classify and extract non-LTRs from RepeatModeler2 library. Additionally, EDTA utilizes AnnoSINE [5] to identify more SINE elements. To equip HiTE with the capability for *de novo* identification of non-LTRs, we have developed a dedicated module named HiTE-NonLTR. It begins by using the coarse-grained repeats generated by our FMEA algorithm. Subsequently, it searches for structural characteristics such as TSDs (8-20 bp) and polyA tails (exceeding 6 bp). Candidate non-LTR elements are then

aligned to the genome to obtain their full-length copies, followed by multiple sequence alignment of all copies. Next, we have designed a dynamic boundary adjustment method to search for the homologous boundaries of the copies, determining the raw 5' and 3' ends. Each copy is then examined to identify the polyA tail near the raw 3' end, establishing the true 3' end of each copy. With the 3' end established, all possible 3' end TSDs are obtained, and corresponding 5' end TSDs are searched near the raw 5' end to determine the true 5' end of each copy. To minimize false positives, we require that a genuine non-LTR element must have a copy count with TSDs greater than half of all copy counts, or exceed 5. While this stringent filtering method ensures reliability in identification, it may result in the loss of some intact LINE elements with low copy numbers. Considering the conservative nature of domains across various TEs, we use a curated LINE domain library to retain candidate LINE elements harboring intact domains. The curated LINE domain library is derived from the *RepeatPeps.lib* within RepeatMasker, which is also used in the classification module of RepeatModeler2.

We have evaluated the performance of three *de novo* methods: RepeatModeler2, EDTA, and HiTE-NonLTR, alongside our homology-based search method, HiTE-NonLTR-homology, using three different evaluation methods: BM_RM2, BM_EDTA, and BM_HiTE. Due to the integration of HiTE-NonLTR and HiTE-NonLTR-homology in HiTE, we have also included evaluations for the merged outcome. As shown in Table R4, according to the results using BM_RM2, HiTE-NonLTR-homology demonstrates superior performance in the majority of species, with higher counts of perfect families compared to three *de novo* methods. At the same time, the perfect families in HiTE do not increase compared to HiTE-NonLTR-homology, indicating that the perfect families identified by HiTE-NonLTR are all included in the known non-LTR library. However, HiTE-NonLTR identifies the highest number of perfect families on *Mouse*, outperforming other methods. This indicates that HiTE-NonLTR is capable of identifying transposons not included in the known non-LTR library. Since the difference between EDTA and RepeatModeler2 lies only in the classifiers used to extract non-LTR elements, they exhibit similar performance. It is worth noting that the number of non-LTR consensus sequences of *Chicken* and *Zebra finch* in Repbase is only 1 and 8, respectively. Consequently, the performance of all tools on these two species is relatively low. Based on the results using BM_EDTA, RepeatModeler2 exhibits higher performance than other methods in species such as *Chicken* and *Zebrafish*. RepeatModeler2 identifies TEs based on sequence repeat characteristics, often detecting fragmented TE families rather than full-length ones [6]. Since BM_EDTA assesses performance at the base level, where partial alignments contribute to true positives, fragmented TE families identified by RepeatModeler2 can achieve higher performance.

In contrast to BM_EDTA, BM_HiTE eliminates the influence from fragments, thus offering a more accurate representation of the true quality of the full-length TE library. BM_HiTE suggests the superior performance of HiTE-NonLTR-homology in *Arabidopsis*, *Rice*, and *Drosophila*. However, when the non-LTRs are not included in the homology library, HiTE-NonLTR-homology exhibits poor performance. For example, in the case of *Mouse*, the *de novo* approach HiTE-NonLTR demonstrates superior performance compared to other methods, such as HiTE-NonLTR-homology. Upon examining the non-LTR homology library used by HiTE-NonLTR-homology, it is evident that there are no non-LTR sequences from *Mouse*.

In summary, we have developed the HiTE-NonLTR module to achieve *de novo* detection of non-LTR elements, demonstrating superior performance compared to existing methods. As shown in Supplementary Table 1, the proportion of non-LTRs in *Mouse* and *Drosophila* is more abundant, which may drive the superior performance of HiTE-NonLTR. However, in species with low non-LTR content such as *Arabidopsis* and *Rice*, existing *de novo* methods still fall short of achieving high-precision performance, which has not yet met the requirements of the High-precision TE Annotator (HiTE). Therefore, HiTE uses both HiTE-NonLTR and HiTE-NonLTR-homology to identify reliable non-LTRs.

We have added detailed descriptions in the '*The identification of non-LTR elements*' section and

Supplementary Note 6 of the revised manuscript, and revised the data in Supplementary Tables 2-4 and Supplementary Tables 9-11 to include evaluations encompassing a broader range of species.

Table R4 Performance evaluation of different non-LTR identification methods. RM2: RepeatModeler2; HiTE-NL: HiTE-NonLTR; HiTE-HO: HiTE-NonLTR-homology; Pe: Perfect; Go: Good; Pr: Present; NF: Not_found; Se: Sensitivity; Pre: Precision; F1: F1-score.

Species	Methods	BM_RM2 (coverage>0.95)				BM_EDTA			BM_HiTE (coverage>0.95)		
		Pe	Go	Pr	NF	Se	Pre	F1	Se	Pre	F1
Arabidopsis	RM2	1	0	4	155	0.6492	0.6025	0.6250	0.0196	0.0424	0.0268
	EDTA	1	0	5	154	0.6185	0.5616	0.5886	0.0159	0.0322	0.0213
	HiTE-NL	3	1	1	155	0.3916	0.9919	0.5615	0.0628	0.9098	0.1175
	HiTE-HO	18	0	3	139	0.6877	0.9779	0.8075	0.1707	0.7683	0.2793
	HiTE	18	1	2	139	0.6946	0.9734	0.8107	0.2128	0.9003	0.3442
Rice	RM2	2	2	14	174	0.6629	0.7981	0.7243	0.0145	0.0202	0.0168
	EDTA	1	5	20	166	0.7196	0.3873	0.5036	0.0236	0.0231	0.0234
	HiTE-NL	9	0	3	180	0.4824	0.5487	0.5134	0.1175	0.0405	0.0602
	HiTE-HO	77	2	16	97	0.7548	0.7580	0.7564	0.7286	0.8546	0.7866
	HiTE	77	3	19	93	0.7602	0.5666	0.6493	0.7176	0.2415	0.3613
Drosophila	RM2	3	4	6	32	0.8754	0.8471	0.8610	0.6222	0.3709	0.4648
	EDTA	3	4	5	33	0.8723	0.8427	0.8572	0.6063	0.2592	0.3631
	HiTE-NL	9	0	1	35	0.6090	0.9953	0.7556	0.8430	0.7222	0.7779
	HiTE-HO	18	2	4	21	0.8453	0.9275	0.8845	0.8903	0.6992	0.7833
	HiTE	18	2	4	21	0.8460	0.9251	0.8838	0.9232	0.6663	0.7740
Chicken	RM2	0	0	0	8	0.9752	0.7643	0.8570	0	0	0
	EDTA	0	0	0	8	0.9826	0.6694	0.7963	0	0	0
	HiTE-NL	0	0	0	8	0.2981	0.4650	0.3633	0	0	0
	HiTE-HO	0	0	2	6	0.2981	0.2692	0.2829	0.0616	0.0008	0.0017
	HiTE	0	0	2	6	0.2981	0.2675	0.2819	0.0616	0.0008	0.0017
Zebra finch	RM2	0	0	1	0	0.9652	0.0023	0.0047	0.3587	0.0023	0.0046
	EDTA	0	0	1	0	0.9657	0.0024	0.0048	0	0	0
	HiTE-NL	0	0	0	1	0	0	0	0	0	0
	HiTE-HO	1	0	0	0	0.0788	0.0002	0.0005	1.0	0.0045	0.0090
	HiTE	1	0	0	0	0.0788	0.0002	0.0005	1.0	0.0045	0.0090
Zebrafish	RM2	4	3	13	414	0.7254	0.6627	0.6927	0.0934	0.2468	0.1356
	EDTA	3	4	11	416	0.7209	0.6630	0.6908	0.3217	0.1756	0.2272
	HiTE-NL	38	4	7	385	0.4094	0.4352	0.4219	0.1609	0.0822	0.1088
	HiTE-HO	191	9	48	186	0.8946	0.5089	0.6487	0.9209	0.0623	0.1167
	HiTE	191	10	47	186	0.8950	0.4486	0.5977	0.9258	0.0624	0.1169
Mouse	RM2	2	1	9	48	0.9489	0.6518	0.7727	0.4583	0.4680	0.4631
	EDTA	2	1	10	47	0.9450	0.6798	0.7908	0.4635	0.4721	0.4678
	HiTE-NL	10	6	11	33	0.3876	0.3695	0.3783	0.9924	0.6535	0.7880

HiTE-HO	0	5	27	28	0.3876	0.3663	0.3766	0.2164	0.0099	0.0190
HiTE	10	9	16	25	0.3876	0.3391	0.3618	0.9927	0.4784	0.6456

References

- [1] Gozashti L and Hosekstra H.E. (2024), “Accounting for diverse transposable element landscapes is key to developing and evaluating accurate de novo annotation strategies” *Genome Biology* 25:4 <https://doi.org/10.1186/s13059-023-03118-1>
- [2] Ou S., Jiang N, Hirsch C.N., and Hufford M.B. (2024), “Response to Commentary: Accounting for diverse transposable element landscapes is key to developing and evaluating accurate de novo annotation strategies” *Genome Biology* 25:6 <https://doi.org/10.1186/s13059-023-03119-0>
- [3] Osmanski, A. B., Paulat, N. S., Korstian, J., Grimshaw, J. R., Halsey, M., Sullivan, K. A., ... & Ray, D. A. (2023). Insights into mammalian TE diversity through the curation of 248 genome assemblies. *Science*, 380(6643), eabn1430.
- [4] Zhang, R. G., Li, G. Y., Wang, X. L., Dainat, J., Wang, Z. X., Ou, S., & Ma, Y. (2022). TESorter: an accurate and fast method to classify LTR-retrotransposons in plant genomes. *Horticulture Research*, 9, uhac017.
- [5] Li, Y., Jiang, N., & Sun, Y. (2022). AnnoSINE: a short interspersed nuclear elements annotation tool for plant genomes. *Plant Physiology*, 188(2), 955-970.
- [6] Storer, J., Hubley, R., Rosen, J., Wheeler, T. J., & Smit, A. F. (2021). The Dfam community resource of transposable element families, sequence models, and genome annotations. *Mobile DNA*, 12, 1-14.

Comment 4. *For the pre-masking step, it is described as finding TEs in one chunk of the genome, and then using the identified TEs to mask the next selected chunk, and so on and so forth until the genome is fully analyzed. The use of this step results in decreased perfect TE families found, sensitivity and precision. So the increase in speed comes at a loss of high quality results. Given that HiTE cannot fully process larger genome without the pre-masking step, I would recommend that this is highlighted in the discussion. While the addition of the pre-masking step does increase the speed of the program, the overall performance of the program decreases. It therefore follows that the program cannot simultaneously be both fast and accurate.*

Authors' Response. Thank you for your comment and suggestion. In the previous version of pre-masking step in HiTE, we used RepeatMasker to mask identified TEs in the next selected chunk. RepeatMasker masks any sequences with high homology to the identified TEs, including both full-length and fragmented sequences. While this approach reduces the running time, it also may lead to the loss of diverse TEs. To address this issue, in the latest HiTE, we use BLASTN [1] for aligning and masking only full-length copies of identified TEs. We use our FMEA algorithm to connect fragmented alignments and retrieve full-length copies (Supplementary Fig. 2b and Supplementary Note 5).

Due to space constraints, we present comparisons of HiTE with existing tools in terms of performance and runtime across *Rice*, *Zebrafish*, *Maize* and *Mouse* (Table R5). For a comprehensive comparison, please refer to Supplementary Tables 2, 4, and 13. The results indicate that HiTE outperforms existing tools in obtaining more perfect families using BM_RM2. When using BM_HiTE, we observe that EarlGrey shows superior performance on *Zebrafish* and *Mouse*, yet exhibits lower performance on other species (Table R5 and Supplementary Table 4). In contrast, HiTE shows higher performance across the majority of species,

demonstrating its reliability and generalization capability. Additionally, when handling large genomes, HiTE requires less runtime compared to existing tools. The previous version of HiTE requires 2.2, 16.5, and 70.5 hours to process the genomes of *Rice*, *Zebrafish*, and *Maize*, whereas in the latest version, HiTE only needs 2, 12, and 25.5 hours, respectively.

We have reassessed the performance of HiTE and updated data into both the main text and appendix, including Figs. 2 and 4, as well as Supplementary Tables 2-4 and 12-13.

Table R5 Performance evaluation of general-purpose TE annotators on *rice*, *zebrafish*, and *maize*, and *mouse*. RM2: RepeatModeler2; Pe: Perfect; Go: Good; Pr: Present; NF: Not_found; Se: Sensitivity; Pre: Precision; F1: F1-score.

Species	Methods	BM_RM2 (coverage>0.95)				BM_HiTE (coverage>0.95)			Running Time (h:m:s)
		Pe	Go	Pr	NF	Se	Pre	F1	
Rice	EDTA	508	271	879	1,390	0.8873	0.8599	0.8734	10:13:31
	RM2	378	104	366	2,200	0.5315	0.5661	0.5482	25:50:58
	EarlGrey	276	247	284	2,241	0.6492	0.5041	0.5675	42:48:26
	HiTE	1,078	100	395	1,475	0.9800	0.8951	0.9356	02:02:54
Zebrafish	EDTA	476	254	713	839	0.6374	0.7720	0.6983	60:13:59
	RM2	483	147	290	1,362	0.5243	0.5728	0.5475	34:30:04
	EarlGrey	509	331	247	1,195	0.9067	0.8239	0.8633	91:26:38
	HiTE	1,120	139	237	786	0.9713	0.7671	0.8572	12:06:27
Maize	EDTA	303	115	396	347	0.8865	0.7477	0.8112	110:56:02
	RM2	341	86	233	501	0.4658	0.4250	0.4445	24:10:44
	EarlGrey	48	172	101	840	0.1663	0.0653	0.0938	71:54:17
	HiTE	446	82	335	298	0.9685	0.8314	0.8947	25:44:09
Mouse	EDTA	34	12	173	338	0.1152	0.2602	0.1597	49:22:36
	RM2	45	17	110	385	0.5396	0.5270	0.5332	16:22:25
	EarlGrey	63	50	133	311	0.9807	0.8232	0.8951	70:27:40
	HiTE	86	20	125	326	0.9855	0.4605	0.6277	10:40:33

References

- [1] BlastN, G. BLAST: basic local alignment search tool. NUTRITIONAL AND PHYSIOLOGICAL DISORDERS IN HORTICULTURAL CROPS (2019).

Comment 5. Please explain the findings in Figure 4c regarding the “Not Found (NF)” category. While the full-length TEs found is generally higher for HiTE than HiTE-FMEA, the NF is also much higher in HiTE. Does this mean that HiTE is missing TE families entirely as compared to Hi-FMEA? Are these novel TEs not captured in the gold standard libraries?

Authors' Response. Thank you for your comment. As shown in Fig. R1b (Fig. 4c in manuscript), compared to HiTE-FMEA, the increase in the “Not Found” count in HiTE is due to a decrease in the counts of “Good” and “Present”. If a gold standard TE sequence is labeled as “Good” or “Present”, it implies that the test library cannot recover the gold standard sequence. Instead, multiple test sequences are needed to cover the gold standard sequence with over 95% coverage, indicating that the test library contains fragments rather than the

full-length TEs (Supplementary Fig. 1b).

It is worth noting that HiTE-FMEA produces a comprehensive set of repetitive sequences, including both TEs and non-TEs. At the same time, the TEs identified within HiTE-FMEA results are typically coarse-grained and fragmented, thus requiring substantial filtering and refinement efforts to obtain a highly reliable TE library. As shown in Fig. R1a (Fig. 4b in manuscript), HiTE-FMEA demonstrates higher sensitivity while maintaining lower precision. The filtering strategy of HiTE is based on structural searches and multiple sequence alignments of TE copies, allowing it to filter out the majority of false positive sequences. Additionally, our dynamic boundary adjustment approach allows the conversion of coarse-grained TEs to fine-grained ones, thereby enabling HiTE to identify more "Perfect" TEs compared to HiTE-FMEA (Fig. R1b).

When the test library contains novel TEs that are not captured in the gold standard libraries, they are likely to be considered as "Good", "Present" or "Not Found". While the outcomes of HiTE-FMEA may include some novel TEs absent in the gold standard libraries, the identification and validation of these novel TEs often pose challenges due to their ambiguous boundaries in multiple sequence alignments of copies. Consequently, the results of HiTE-FMEA are highly valuable, and with ongoing advancements in identification methods, we believe HiTE still holds potential for improvement.

Fig. R1 Evaluation of HiTE against HiTE-FMEA using BM_HiTE (a) and BM_RM2 (b) under rough and precise perfect definitions, respectively.

Comment 6. Although counterintuitive, using the “-nolow” parameter as part of RepeatMasker actually causes the TEs in the library to map to simple repeats and low complexity regions. As part of running RepeatMasker, the first and last stages are to search for young and old simple repeats, respectively. As some consensus sequences within a library, for example, Alu, contain within them simple repeat sequences, such as polyA tails and low complexity regions (GC-rich regions of Alu and LINE as examples). If these low complexity and simple repeat regions are not masked, this leads to an increase in false positives. In addition,

the usage of the “-species” parameter vs. “-lib” is not specific to Dfam libraries. However, given that the same parameter was used for all species, my point is moot. While the same setting was used for all benchmarking purposes, the “-q” setting in RepeatMasker leads to less sensitive searches as compared to the default setting, or the “-s” setting, and may influence your results.

Authors' Response. We thank the reviewer for the detailed and insightful explanation. In the benchmarking methods of RepeatModeler2 (BM_RM2) and EDTA (BM_EDTA), their RepeatMasker commands are as follows: "RepeatMasker -lib rebase_lib -nolow -pa threads test_lib" and "RepeatMasker -e ncbi -pa threads -q -no_is -norna -nolow -div 40 -lib [rebase_lib/test_lib] -cutoff 225 assembly". To ensure the reproducibility of the experiments, we adhere to the identical RepeatMasker parameters as advised by the original authors. Regarding the benchmarking method of HiTE (BM_HiTE), we previously used RepeatMasker to align the Rebase and test library to the genome, preserving their respective full-length copies. Subsequently, we computed TP, FP, and FN by assessing the differences between these full-length copies. We considered a match as a true positive only if the length of the overlap between the test and Rebase full-length TE sequence exceeded 95% of their lengths.

In the latest BM_HiTE, we have replaced RepeatMasker with BLASTN as the alignment tool and accordingly modified the method for obtaining full-length copies of TE consensus sequences. Previously, we parsed the .out alignment files generated by RepeatMasker to retrieve full-length copies that cover more than 95% of the TE consensus sequence length. Now, we rely on BLASTN alignment results and use our *de novo* TE searching algorithm, FMEA, to assemble fragmented alignments. Our FMEA algorithm can bridge alignment gaps caused by divergences between TE copies, allowing the retrieval of TE copies with certain differences from the TE consensus sequence. We have opted to use BLASTN along with our FMEA algorithm instead of RepeatMasker, as outlined below:

Although RepeatMasker supports the use of the "-div" parameter to mask divergent repeats with TE consensus, it does not provide a parameter to skip over alignment gaps. To achieve finer control over the differences among TE copies, we have taken into account the divergence between TE instances when retrieving full-length copies (Supplementary Fig. 2b). To obtain full-length copies of TEs, we use our FMEA approach to concatenate BLASTN alignment results, setting the maximum gap threshold for fragment concatenation to $TE_length * full_length_coverage$, where TE_length represents the length of the TE consensus sequence, and $full_length_coverage$ denotes the coverage threshold for full-length copies set by the user in BM_HiTE. Through our testing, BM_HiTE has been confirmed to effectively assess the true performance of different tools (Supplementary Table 4).

We have revised all data involving BM_HiTE in both the main text and the supplementary materials, including Figs. 2-4, Supplementary Figs. 2 and 11, Supplementary Note 2, and Supplementary Tables 4, 7, 8, 11, 12, 14.

Answers to Reviewer #3

Reviewer #3 (Remarks to the Author):

Comment 1. *I find that the explanations or analyses provided by most authors are reasonable. However, I have reservations regarding the requirement for TE annotation evaluation to be >95% identity/coverage threshold for. This evaluation actually comes from two aspects: firstly, the examination of the library itself. It is reasonable to impose stricter parameters when comparing curated libraries with de novo annotated libraries. However, since the library itself is derived from a consensus step, traditional rules such as 80-80-80 (80% identity, 80% coverage and > 80 bp) apply already introduce the difference even in the same sequence. Therefore, even if they are both full-length, there may still be some differences between them. Additionally, different clustering and filtering for nested TEs also break down the full-length TE candidates. A more significant issue arises from the uncertainty of annotation. When annotating the full-length library with Repeatmasker, all fragmented TEs can be annotated. Thus, requiring 95% coverage/identity is unreasonable. For instance, soloLTRs, which only consist of a portion of the full-length, would be excessively constrained. The authors could filter out the highly fragmented results from Repeatmasker's output, but the requirement for coverage and identity conflicts with the essence of homology-based annotation.*

Authors' Response. We thank the reviewer for the insightful comments. We fully agree with the reviewer that “*since the library itself is derived from a consensus step, traditional rules such as 80-80-80 (80% identity, 80% coverage and > 80 bp) apply already introduce the difference even in the same sequence. Therefore, even if they are both full-length, there may still be some differences between them.*”. In steps 1 and 2 of BM_HiTE, the standard library and test library are aligned to the genome to obtain full-length copies of TE consensus sequences. In previous implementations, we used RepeatMasker for this purpose. To achieve finer control over the differences among TE copies, in the latest version, we use BLASTN and our FMEA algorithm to obtain full-length copies (Supplementary Fig. 2). Therefore, even though different TE identification methods use varying thresholds to generate TE consensus, BM_HiTE can capture full-length copies on the genome that exhibit certain variations.

Due to divergence among TE instances, BLASTN alignments often produce fragmented matches (Supplementary Fig. 2b). To address this issue, we use our FMEA algorithm to obtain full-length copies, bridging alignment gaps caused by differences. We set the maximum gap threshold for fragment concatenation to $TE_length * full_length_coverage$, where TE_length represents the length of the TE consensus sequence and $full_length_coverage$ denotes the coverage threshold for full-length copies set by the user in BM_HiTE, for example, 0.8. This approach enables us to effectively identify copies even when there are differences between TE consensus sequences and their copies. Additionally, for LTR retrotransposons, tools like EDTA, RepeatModeler2, and HiTE separate their internal and terminal sequences. In BM_HiTE, we treat LTR terminal and internal sequences as separate elements, allowing for effective evaluation of solo-LTRs as well.

Given that different tools may generate TE consensus sequences using varying methods, applying a 95% or 99% coverage threshold may not be universally applicable. Thus, we support user-defined threshold values in BM_HiTE. Besides testing with 95% and 99% coverage thresholds, we also introduce an 80% threshold, aligning with the traditional 80-80-80 rule for consensus sequence generation. As shown in Table R6, HiTE demonstrates superior performance under all thresholds across nearly all species except for *Mouse (M. musculus)* when compared to RepeatScout, EDTA, RepeatModeler2, and EarlGrey. Although HiTE achieves the highest performance at the 80% coverage threshold on *Mouse*, EarlGrey obtains higher performance at 95% and 99% thresholds. However, due to the unstable performance of EarlGrey across different species and its longer runtime

(Supplementary Table 14), it cannot meet the increasing demand for TE identification and annotation in non-model species.

We have added an 80% threshold of BM_HiTE for testing, and revised all data involving BM_HiTE in both the main text and the supplementary materials, including Figs. 2-4, Supplementary Figs. 2 and 11, Supplementary Note 2, and Supplementary Tables 4, 7, 8, 11, 12, 14.

Table R6 Performance evaluation of general-purpose TE annotators using different genomes based on the benchmarking method of HiTE (BM_HiTE). Se: Sensitivity; Pre: Precision; F1: F1-score; NA: Not applicable.

Species	Tools	BM_HiTE (coverage>0.8)			BM_HiTE (coverage>0.95)			BM_HiTE (coverage>0.99)		
		Sen	Pre	F1	Sen	Pre	F1	Sen	Pre	F1
C. briggsae	RepeatScout	0.7083	0.2043	0.3172	0.6188	0.2222	0.3270	0.4255	0.1711	0.2440
	EDTA (sensitive)	0.1981	0.1923	0.1952	0.0800	0.0888	0.0841	0.0576	0.0642	0.0607
	RepeatModeler2	0.6519	0.3252	0.4339	0.4755	0.2438	0.3223	0.3126	0.1977	0.2422
	EarlGrey	0.9039	0.5451	0.6801	0.8448	0.6454	0.7318	0.5089	0.5101	0.5095
	HiTE	0.8391	0.6378	0.7247	0.8079	0.6407	0.7147	0.7455	0.5931	0.6606
A. thaliana	RepeatScout	0.2575	0.1664	0.2022	0.0906	0.0720	0.0802	0.0808	0.0668	0.0731
	EDTA (sensitive)	0.8191	0.8981	0.8568	0.8303	0.8970	0.8624	0.8433	0.8933	0.8676
	RepeatModeler2	0.4116	0.4611	0.4349	0.1361	0.1988	0.1616	0.1144	0.1891	0.1425
	EarlGrey	0.6542	0.4615	0.5412	0.2346	0.1903	0.2101	0.1159	0.1059	0.1107
	HiTE	0.9016	0.9589	0.9294	0.9125	0.9582	0.9348	0.9169	0.9547	0.9354
D. melanogaster	RepeatScout	0.5465	0.0820	0.1426	0.2112	0.0358	0.0613	0.2068	0.0426	0.0707
	EDTA (sensitive)	0.3673	0.2165	0.2724	0.2921	0.1918	0.2316	0.2539	0.1735	0.2061
	RepeatModeler2	0.7252	0.2174	0.3346	0.6775	0.2336	0.3475	0.6294	0.2191	0.3251
	EarlGrey	0.8788	0.4684	0.6111	0.5263	0.1700	0.2569	0.3864	0.1213	0.1846
	HiTE	0.8041	0.6904	0.7429	0.7130	0.5592	0.6268	0.7072	0.5388	0.6116
O. sativa	RepeatScout	0.6460	0.4416	0.5246	0.3938	0.3038	0.3430	0.3277	0.2908	0.3082
	EDTA (sensitive)	0.9363	0.8968	0.9161	0.8873	0.8599	0.8734	0.8908	0.8555	0.8728
	RepeatModeler2	0.7381	0.7099	0.7237	0.5315	0.5661	0.5482	0.4729	0.5279	0.4989
	EarlGrey	0.8300	0.7036	0.7616	0.6492	0.5041	0.5675	0.5543	0.4598	0.5026
	HiTE	0.9876	0.9182	0.9516	0.9800	0.8951	0.9356	0.9737	0.8816	0.9253
D. rerio	RepeatScout	NA	NA	NA	NA	NA	NA	NA	NA	NA
	EDTA (sensitive)	0.7491	0.7838	0.7661	0.6374	0.7720	0.6983	0.6026	0.7727	0.6771
	RepeatModeler2	0.6444	0.6803	0.6619	0.5243	0.5728	0.5475	0.3900	0.4573	0.4210
	EarlGrey	0.9123	0.7483	0.8222	0.9067	0.8239	0.8633	0.6982	0.7466	0.7216
	HiTE	0.9670	0.7769	0.8616	0.9713	0.7671	0.8572	0.9564	0.7072	0.8131
Z. mays	RepeatScout	NA	NA	NA	NA	NA	NA	NA	NA	NA
	EDTA (sensitive)	0.9370	0.7625	0.8408	0.8865	0.7477	0.8112	0.8802	0.7968	0.8364

	RepeatModeler2	0.7321	0.6439	0.6852	0.4658	0.4250	0.4445	0.1970	0.1828	0.1896
	EarlGrey	0.9510	0.6683	0.7850	0.1663	0.0653	0.0938	0.1204	0.0717	0.0899
	HiTE	0.9712	0.8244	0.8918	0.9685	0.8314	0.8947	0.9685	0.8429	0.9013
	RepeatScout	NA	NA	NA	NA	NA	NA	NA	NA	NA
	EDTA (sensitive)	0.3883	0.1113	0.1730	0.4373	0.1055	0.1700	0.5035	0.1009	0.1681
G. gallus	RepeatModeler2	0.5264	0.1184	0.1933	0.5421	0.1319	0.2121	0.5475	0.1290	0.2089
	EarlGrey	0.7876	0.0538	0.1008	0.6228	0.0340	0.0645	0.5077	0.0240	0.0459
	HiTE	0.8577	0.2366	0.3709	0.8833	0.2701	0.4137	0.8869	0.2986	0.4467
	RepeatScout	NA	NA	NA	NA	NA	NA	NA	NA	NA
	EDTA (sensitive)	0.5472	0.2837	0.3737	0.4823	0.2568	0.3351	0.3784	0.2090	0.2693
T. guttata	RepeatModeler2	0.8550	0.3760	0.5223	0.6522	0.2941	0.4054	0.5219	0.2451	0.3335
	EarlGrey	0.8247	0.2722	0.4093	0.7483	0.2393	0.3627	0.6260	0.1886	0.2898
	HiTE	0.7283	0.4768	0.5763	0.6512	0.4811	0.5534	0.5981	0.4984	0.5437
	RepeatScout	NA	NA	NA	NA	NA	NA	NA	NA	NA
	EDTA (sensitive)	0.0527	0.4047	0.0932	0.1152	0.2602	0.1597	0.0802	0.1541	0.1055
M. musculus	RepeatModeler2	0.9586	0.9695	0.9641	0.5396	0.5270	0.5332	0.2857	0.3575	0.3176
	EarlGrey	0.9962	0.9092	0.9507	0.9807	0.8232	0.8951	0.8629	0.6743	0.7570
	HiTE	0.9998	0.9882	0.9940	0.9855	0.4605	0.6277	0.8460	0.1292	0.2242

Comment 2. Another concern arises from the premature masking of tandem repeats, which may result in the loss of many LTRs, such as the Dasheng LTR in rice (Jiang et.al 2002), which contains numerous internal tandem repeats. With the proliferation of T2T genomes, an increasing number of complex regions are being assembled, such as the centromere-specific ATHILA family in *Arabidopsis thaliana* (Naish et.al 2021) and CRM in maize. Pre-masking could also lead to missed detections of such elements although increase the speed. Can we resecure these LTR in these masking regions? Or identify them before masking the whole genome. Given that the genomes evaluated by the authors were previously assembled incompletely, it is uncertain how this step would impact organisms with T2T genomes, such as rice, *Arabidopsis*, or humans. Could the authors consider evaluating the extent of this impact in the discussion or as a supplement?

Authors' Response. Thank you for the comment and suggestion. This issue is indeed very important and is the one that we had overlooked previously. We are providing point-by-point responses to your comment:

HiTE is currently unable to identify LTRs with large tandem repeats: Typically, tandem repeats are masked before conducting TE identification and annotation, which can eliminate false positives caused by tandem repeats and reduce runtime. However, the detection of LTR transposons in HiTE uses the genome without pre-masking tandem repeats, as shown in Fig. 1h of the main text. Our current LTR identification module is based on a pipeline composed of LTR_harvest, LTR_finder, and LTR_retriever, which has been applied to multiple tools, including EDTA and RepeatModeler2.

The Dasheng LTR in rice is a long non-autonomous transposon that does not encode proteins [1]. Our analysis reveals that while this transposon is identified by LTR_finder, it is filtered out by LTR_retriever, which currently

filters out LTR elements containing large tandem repeats. Therefore, this constraint will be introduced by any TE identification tool using LTR_retriever.

HiTE can identify the TE families within the centromeric/telomeric regions: Naish et al. generated a T2T Arabidopsis thaliana genome and observed multiple insertions of the ATHILA family within the centromeric tandem repeat region CEN180 (Fig. R2). Referring to Table S6 in the literature [2], we pinpointed the ATHILA5 transposon positioned at *Chr5: 12739578-12750538*. As shown in Fig. R3A, HiTE successfully identifies this transposon, as CEN180 is only present in a small portion of the 5' LTR (Fig. R2B). Similarly, we examined CRM in maize [3], which itself does not harbor large tandem repeats. Consequently, HiTE can also detect CRM transposons (Fig. R3B).

Therefore, although LTR_retriever filters out LTRs containing internal large tandem repeats, it can identify elements like ATHILA and CRM. These elements, though inserted into the telomeres or centromeres, do not possess large tandem repeats themselves.

Fig. R2 Higher-order duplication of ATHILA elements post-integration (Figure S11 from reference [2]).

Fig. R3 The screenshot demonstrates that HiTE can identify ATHILA (A) and CRM (B) elements inserted into the centromeric high-order repeat.

HiTE can identify other types of TEs containing large tandem repeats: In addition to LTRs, we also

examined other TE identification modules within HiTE and found that HiTE can identify transposons containing large tandem repeats. As shown in Fig. 1d of the manuscript, following pre-masking of tandem repeats, we have developed a *de novo* TE searching method named HiTE-FMEA, which enables the bridging of large gaps in alignments. Since our *de novo* identification of TIR, Helitron, and non-LTR transposons uses the coarse-grained TE results identified by HiTE-FMEA, HiTE can identify TIR, Helitron, and non-LTR transposons containing internal tandem repeats. Fig. R4 depicts a scenario where an 11 kb TIR element is situated in rice, located at *chr1:3748476-3759610*, harboring large tandem repeats internally. Using the HiTE-FMEA algorithm, HiTE can identify this full-length transposon.

Fig. R4 HiTE identifies long TIR elements containing tandem repeats. Annotations for Rebase (in burgundy, directional) and TRF (in black, non-directional) are obtained from the UCSC browser, while directional annotations in black are generated by HiTE. It can be observed that the SPMLIKE elements in Rebase contain numerous tandem repeats. Despite this, HiTE is still able to accurately identify them, obtaining the reverse-complemented sequences with Rebase.

Fig. R5 The illustration depicts that the HiTE-FMEA algorithm can skip the gaps caused by masking tandem repeats.

The HiTE-FMEA algorithm can identify internal regions containing a large number of tandem repeats as described below: illustrated in Fig. R5, the algorithm handles a transposon with internal tandem repeats by initially pre-masking these repeats. The tandem repeat segment of the transposon is designated as N, while the

remainder remains unchanged. When using such sequences for self-alignment to detect TE candidates, multiple fragmented alignments occur. HiTE facilitates this process by supporting user-defined joining gap parameters "--fixed_extend_base_threshold", with a default setting of 1000 bp. This parameter aids in the recovery of multiple fragmented alignments, consolidating them into full-length TE candidates. For further details on the FMEA algorithm, please refer to Supplementary Note 5.

Assessing the impact of T2T genome on HiTE in comparison to genomes with gaps: Compared to existing genomes, T2T genomes typically address the assembly of complex regions such as telomeres and centromeres. We used T2T assemblies of *Arabidopsis thaliana* [2] and *rice* [4] to evaluate the influence of employing T2T assemblies on HiTE performance. As shown in Table R7, although T2T assemblies do not exhibit a significant increase in the number of perfect families identified in HiTE compared to assemblies with gaps, they do show notable performance improvements in BM_HiTE across different threshold evaluations, particularly with stricter thresholds like 95% and 99%. This indicates that the use of T2T assemblies can yield favorable TE annotation results.

We have added this analysis in *Discussion*, Supplementary Note 10, and Supplementary Table 12 of the revised manuscript.

Table R7 The impact of different assemblies on the performance of HiTE. *Arab*: *Arabidopsis*; *Pe*: Perfect; *Go*: Good; *Pr*: Present; *NF*: Not_found; *Se*: Sensitivity; *Pre*: Precision; *F1*: F1-score.

Species	Assembly	BM_RM2 (coverage>0.95)				BM_HiTE (coverage>0.8)			BM_HiTE (coverage>0.95)			BM_HiTE (coverage>0.99)		
		Pe	Go	Pr	NF	Se	Pre	F1	Se	Pre	F1	Se	Pre	F1
Arab	Gaps	281	25	57	557	0.9193	0.9659	0.9420	0.9053	0.9440	0.9242	0.8937	0.9264	0.9098
	T2T	290	24	56	550	0.9394	0.9678	0.9534	0.9476	0.9627	0.9551	0.9508	0.9600	0.9554
Rice	Gaps	1,089	110	382	1,467	0.9848	0.9264	0.9547	0.9727	0.8711	0.9191	0.9584	0.8152	0.8810
	T2T	1,083	107	389	1,469	0.9890	0.9306	0.9589	0.9832	0.9161	0.9485	0.9738	0.8825	0.9259

References

- [1] Jiang, N., Bao, Z., Temnykh, S., Cheng, Z., Jiang, J., Wing, R. A., ... & Wessler, S. R. (2002). Dasheng: a recently amplified nonautonomous long terminal repeat element that is a major component of pericentromeric regions in rice. *Genetics*, 161(3), 1293-1305.
- [2] Naish, M., Alonge, M., Wlodzimierz, P., Tock, A. J., Abramson, B. W., Schmücker, A., ... & Henderson, I. R. (2021). The genetic and epigenetic landscape of the *Arabidopsis* centromeres. *Science*, 374(6569), eabi7489.
- [3] Chen, J., Wang, Z., Tan, K., Huang, W., Shi, J., Li, T., ... & Lai, J. (2023). A complete telomere-to-telomere assembly of the maize genome. *Nature genetics*, 55(7), 1221-1231.
- [4] Shang, L., He, W., Wang, T., Yang, Y., Xu, Q., Zhao, X., ... & Qian, Q. (2023). A complete assembly of the rice Nipponbare reference genome. *Molecular Plant*, 16(8), 1232-1236.

Comment 3. Thanks for your improvements on the documentation and distribution software. But I find the result still a bit messy for users. For instance, HiTE try to find full-length TE than other tools but only provides fasta. For TE biology, detailed structure of TE could be very helpful for people manually curation and comparing. Do the authors think provide a full-length TE annotation bed or gff? Like TSD, 3'LTR,5'LTR, terminal inverted repeats, hairpin loops, and polyA/T tails.

Authors' Response. Thank you for the suggestion. We have improved HiTE by adding the "--intact_anno" parameter. Users can set it to 1 to generate a full-length TE annotation file in gff format. As shown in Fig. R6, the file includes information such as TSD, 3'-LTR, 5'-LTR, terminal inverted repeats, hairpin loops, and polyA/T tails.

```

Chr1 HiTE repeat_region 2535673 2539068 . - . id=repeat_region_15:name=Chr1:2535678..2539063;classification=LTR/Copia;ltr_identity=1.0000;motif=TCGA;tsd=GAACA
Chr1 HiTE target_site_duplication 2535673 2535677 . - . id=LTR_15;parent=repeat_region_15:name=Chr1:2535678..2539063;classification=LTR/Copia;ltr_identity=1.0000;motif=TCGA;tsd=GAACA
Chr1 HiTE long_terminal_repeat 2535678 2535812 . - . id=LTR_15;parent=repeat_region_15:name=Chr1:2535678..2539063;classification=LTR/Copia;ltr_identity=1.0000;motif=TCGA;tsd=GAACA
Chr1 HiTE LTR 2535678 2539063 . - . id=LTRRT_15;parent=repeat_region_15:name=Chr1:2535678..2539063;classification=LTR/Copia;ltr_identity=1.0000;motif=TCGA;tsd=GAACA
Chr1 HiTE long_terminal_repeat 2538929 2539063 . - . id=LTR_15;parent=repeat_region_15:name=Chr1:2535678..2539063;classification=LTR/Copia;ltr_identity=1.0000;motif=TCGA;tsd=GAACA
Chr1 HiTE target_site_duplication 2539064 2539068 . - . id=TSD_15;parent=repeat_region_15:name=Chr1:2535678..2539063;classification=LTR/Copia;ltr_identity=1.0000;motif=TCGA;tsd=GAACA
Chr1 HiTE TIR 2539734 2540019 . + . id=te_intact_531:name=TIR_261;classification=DNA/PIF-Harbinger;tir=17,270-286;tir_identity=0.941176;tsd=AA;tsd_len=2
Chr1 HiTE TIR 2555924 2556142 . - . id=te_intact_6050:name=TIR_187;classification=DNA/PIF-Harbinger;tir=1-27,193-210;tir_identity=0.851852;tsd=TAA;tsd_len=3
Chr1 HiTE Helitron 2628215 2628513 . + . id=te_intact_440:name=Helitron_7;classification=RC/Helitron;hairpin_loop=NA
Chr1 HiTE Helitron 2680312 2680956 . + . id=te_intact_8081:name=Helitron_77;classification=RC/Helitron;hairpin_loop=CCCGCAGCAACGCCGCCGG
Chr1 HiTE TIR 2756384 2756695 . - . id=te_intact_8480:name=TIR_170;classification=DNA/PIF-Harbinger;tir=15,298-312;tir_identity=0.8;tsd=TTA;tsd_len=3
Chr1 HiTE TIR 2757057 2757636 . - . id=te_intact_9430:name=TIR_696;classification=DNA/HAT;tir=1-15,539-570;tir_identity=1.0;tsd=GTGGGG;tsd_len=8
Chr1 HiTE TIR 2782462 2782925 . + . id=te_intact_124:name=TIR_538;classification=DNA/PIF-Harbinger;tir=18,455-464;tir_identity=0.9;tsd=TACCAGTGTGA;tsd_len=12
Chr1 HiTE Non_LTR 2787863 2787985 . - . id=te_intact_8352:name=Homology_Non_LTR_27;classification=SINE/TRNA;polya_t=AAAAAA;tsd=AA;tsd_len=2
Chr1 HiTE TIR 2821747 2821969 . - . id=te_intact_4296:name=TIR_992;classification=DNA/MULE;tir=1-20,204-223;tir_identity=0.75;tsd=CCCCTAGAC;tsd_len=9
Chr1 HiTE TIR 2855353 2855943 . + . id=te_intact_6294:name=TIR_682;classification=DNA/HAT;tir=NA;tsd=ATAGT;tsd_len=6

```

Fig. R6 The screenshots of the HiTE full-length TE annotation file.

Comment 4. Supp Fig9, why EDTA result only contained TIR and LTR? Did the author exclude the Helitron and RepeatModeler running?

Authors' Response. Thank you for your attention to detail. Initially, we used the default parameters to run EDTA, which resulted in excluding the execution of RepeatModeler. To address your concerns, we conducted a re-run of EDTA in sensitive mode and subsequently updated all results associated with EDTA, including Supplementary Fig. 9 and Supplementary Tables 2-4.

In the previous Supplementary Fig. 9, the proportion of Helitron elements evaluated by EDTA across all species was 0. This discrepancy arose from the inconsistent labeling of Helitron tags. In EDTA, Helitron elements are labeled as DNA/Helitron, whereas RepeatMasker uses RC/Helitron. Since we used the .tbl file outputted by RepeatMasker to calculate the proportions of different TEs, the Helitron elements identified by EDTA were erroneously categorized under DNA transposons. Therefore, following the recommendation of EDTA authors, we re-calculated the proportions of different TE types using the buildSummary.pl script in RepeatMasker.

We have updated the results of Supplementary Fig. 9 and Supplementary Tables 2-4 in the revised manuscript.

REVIEWERS' COMMENTS

Reviewer #1 (Remarks to the Author):

Thank you for your thorough responses to all of the reviewers. I appreciate your patience, as it took more time than anticipated to review all of the necessary material, as there were many fruitful analyses and tables added to the manuscript.

I fully support the manuscript in its current form. Thank you for your hard work and dedication to HiTE.

Reviewer #3 (Remarks to the Author):

Thanks for the careful benchmark and improvement on the software! The authors have provided an excellent response to my concerns.